# Seasonality and scenario dependence of rapid Arctic sea ice loss events in CMIP6 simulations

Annelies Sticker[1,*], François Massonnet[1], Thierry Fichefet[1], Patricia DeRepentigny[1], Alexandra Jahn[2,3], David Docquier[4], Christopher Wyburn-Powell[2,3], Daphne Quint[2], Erica Shivers[2], and Makayla Ortiz[2]

[1]Earth and Life Institute, Earth and Climate, UCLouvain, Louvain-la-Neuve, Belgium
[2]Department of Atmospheric and Oceanic Sciences, University of Colorado Boulder, Boulder, CO, USA
[3]Institute for Arctic and Alpine Research, University of Colorado Boulder, Boulder, CO, USA
[4]Royal Meteorological Institute of Belgium, Brussels, Belgium
[*]Corresponding author: Annelies Sticker (annelies.sticker@uclouvain.be)

**Abstract.**

The end-of-summer Arctic Ocean is projected to face at least one occurrence of practically ice-free conditions (sea ice extent $< 1$ million $km^2$) by the middle of the century under all Coupled Model Intercomparison Project phase 6 (CMIP6) scenarios. Climate models indicate that this transition toward a practically ice-free Arctic Ocean in late summer will be punctuated by rapid ice loss events (RILEs), i.e., year-to-year reductions in total sea ice extent that occur at a much faster rate than expected from the forced contribution. The extreme sea ice loss associated with RILEs in climate models exceeds any observed rates of sea ice loss since the start of the satellite era, including the largest observed rate of $-0.28$ million $km^2$ per year during 2001–2008. As such, there could be a much faster transition toward practically ice-free conditions than expected based on a linear trend of past observations. However, RILEs are not well understood and it is currently impossible to predict their occurrence a season to several years ahead. It is therefore essential to improve our understanding of these events. This study presents the first comprehensive analysis of RILEs in a diverse set of 26 CMIP6 models, including five large ensembles, following both low and high warming scenarios over the period from 1970 to 2100. Our analysis shows that RILEs are expected to occur year-round, but the timing and duration of these events are found to be season-dependent, with less frequent but longer-lived RILEs in winter and spring, and more frequent but shorter-lived RILEs in summer and autumn under a high emission scenario. In addition, we find that the warming scenario has a greater influence on RILE characteristics in the winter/spring season than in summer/autumn. Our results also emphasize that model uncertainty is larger regarding the probability and characteristics of RILEs for winter/spring events compared to summer/autumn ones. Finally, while the initial sea ice extent at which RILEs are triggered depends on whether they occur in September or March, the initial sea ice volume is similar for both months, which emphasizes the critical role of sea ice thickness as a preconditioning factor for RILEs. Based on CMIP6 models, there is an approximately 60% chance that at least one summer RILE starts before 2030 in September. This study of RILEs is particularly opportune as CMIP6 models suggest that, following a period of relative stability in Arctic sea ice, the probability of a rapid sea ice reduction increases. Given the relatively stable conditions observed between 2015–2024, the current summer Arctic sea ice state may have an increased probability to be on the verge of a rapid sea ice loss event.

# 1 Introduction

The state of the sea ice cover in the Arctic stands as an important sign of the region's transition to a warmer climate, highlighting its role both as an indicator and driver of global change (Serreze et al., 2009; Serreze and Barry, 2011; Taylor et al., 2013; Meredith et al., 2019). Over the past few decades, the Arctic has undergone large changes, leaving the sea ice system in a new state (Landrum and Holland, 2020). The sea ice extent (SIE) at the end of summer has diminished by 12.13% per decade between 1979 and 2024 relative to the 1981–2010 average (Fetterer et al., 2017). The decrease in Arctic SIE occurs not only in September but also throughout the year (Onarheim et al., 2018), and in addition to the reduction in extent, the sea ice cover is also much younger and thinner, making the ice that survives year-round more vulnerable to atmospheric and oceanic forcing (Stroeve and Notz, 2018).

The long-term negative trend in Arctic SIE is largely attributed to the increase in greenhouse gas concentrations in the atmosphere (Stroeve and Notz, 2015; Meredith et al., 2019). However, superimposed upon this trend is an interannual to decadal variability leading to periods of relative stability interrupted by abrupt sea ice declines (Kay et al., 2011; Swart et al., 2015; Baxter et al., 2019). As an example, Arctic sea ice retreated more than three times faster in the first decade of the 21$^{\text{st}}$ century (2001–2010: -1.7 million $\text{km}^2$/decade) than it did in the last two decades of the 20$^{\text{th}}$ century (1981–2000: -0.5 million $\text{km}^2$/decade). More recently, the September sea ice extent trend over 2012–2021 has been slightly positive (0.027 million $\text{km}^2$/decade).

Accelerated sea ice retreat during one or over several consecutive seasons can have profound impacts on the Arctic environment. During such periods, accessibility of shipping routes can be greatly enhanced for several months of the year and winter sea ice might become thin enough to let light icebreakers cruise to the Arctic safely all year round (Crawford et al., 2021). Ecosystems can also feel the effects of sudden multi-year sea ice retreats, as the length of the sea ice season exerts a first-order control on the amount of light reaching phytoplankton, the building blocks of the Arctic food web (Arrigo and van Dijken, 2011; Wassmann et al., 2011). Finally, by exposing more of the Arctic Ocean to the atmosphere for several years in a row, extended periods of large sea ice decline can enhance the ice-albedo feedback and lead to increased temperature and evaporation, which could translate to extreme weather events in the terrestrial regions of the Arctic periphery (e.g., Alaska, Svalbard, coastal Siberia; Screen et al., 2015; Delhaye et al., 2023; Lawrence et al., 2008).

Sea ice loss events are also studied on shorter time scales, with Very Rapid Ice Loss Events (VRILEs) describing abrupt declines in sea ice that happen over days to weeks (e.g., Wang et al., 2020; McGraw et al., 2022; Frank, 2024). VRILEs are often associated with atmospheric and oceanic anomalies that enhance ice loss over short periods, typically within a season. While these studies have deepened our understanding of subseasonal sea ice variability, the focus of the present study is on RILEs, which manifest on subdecadal to decadal timescales.

The concept of a RILE was first proposed by Holland et al. (2006) when they identified periods of abrupt reduction in summer Arctic sea ice in seven simulations of the Community Climate System Model (CCSM) version 3. Several modeling studies on RILEs have since followed (e.g., Lawrence et al., 2008; Holland et al., 2008; Döscher and Koenigk, 2013; Paquin et al., 2013; Auclair and Tremblay, 2018; Mioduszewski et al., 2019; Rieke et al., 2023) and they all project that RILEs will

become more prevalent in the upcoming decades as sea ice variability rises. Despite the previous studies about RILEs, we still currently lack a year-round overview of the properties of RILEs and the mechanisms underlying the occurrence of RILEs remain poorly understood, posing challenges in accurately predicting their onset from one season to several years in advance.

The latest generation of models participating in the Coupled Model Intercomparison Project (CMIP6) exhibit several improvements in their representation of global and polar climate. These models show a more realistic estimate of the sensitivity of September Arctic sea ice area to $CO_2$ emissions and improved representation of sea ice dynamics (SIMIP Community, 2020; Watts et al., 2021), making it worthwhile to reassess Arctic sea ice variability through the lens of RILEs. In this study, we present the first investigation of RILEs year-round using a multi-model ensemble gathering 26 different climate models as well as five large ensembles (Sect. 2). The impact of different emission scenarios is also evaluated using two future Shared Socioeconomic Pathways (SSPs): a low-emission scenario (SSP1-2.6) and a high-emission scenario (SSP5-8.5). We first assess the seasonality of RILEs, highlighting large differences in the characteristics of RILEs that occur in the first versus last six months of the year (Sect. 3.1). We also look at the probability of RILEs in CMIP6 simulations (Sect. 3.2). Then we focus on the timing of RILEs as well as the SIE and total sea ice volume (SIV) at which RILEs start, focusing on September and March RILEs (Sect. 3.3). Finally, we discuss the implications of our results and conclude in Sect. 4 and 5.

## 2 Data and Methods

### 2.1 Data

We analyze data from the first ensemble member of 26 CMIP6 models (see Table 1) that were chosen based on the availability of the sea ice variables SIE, sea ice concentration (SIC) and SIV. The nominal horizontal resolution of the ocean/sea ice component from the different CMIP6 models varies between 25 and 250 $km$, with the majority using a resolution of 100 $km$ (Table 1). We use output from historical simulations, which cover the period 1850 to 2014, except for the EC-Earth3 large ensemble spanning from 1970 to 2014. We also employed two sets of climate projections following low and high warming scenarios, specifically SSP1-2.6 and SSP5-8.5, which correspond to a top-of-atmosphere radiative forcing in 2100 of 2.6 and 8.5 $W\,m^{-2}$ with respect to pre-industrial levels, respectively (O'Neill et al., 2016). Under SSP1-2.6, the Arctic SIE continues to decline in the earlier decades of the 21st century before stabilizing towards the latter part of the century (Fig. S1). The climate projections cover the period 2015 to 2100, except for the CAMS-CSM1-0 model (model #4 in Table 1) which ranges from 2015 to 2099. We only use the years covered by all models and, as such, our study focuses on the time period 1970–2099.

In addition to the CMIP6 multi-model ensemble that allows for a detailed investigation of model uncertainty, we also analyze historical and SSP5-8.5 simulations from five large ensembles to better understand the role of internal climate variability on our results: ACCESS-ESM1.5 (40 ensemble members; Ziehn et al., 2020), CanESM5 (25 ensemble members; Swart et al., 2019d), EC-Earth3 (50 ensemble members; Wyser et al., 2021), MIROC6 (50 ensemble members; Shiogama et al., 2023), and MPI-ESM1.2-LR (30 ensemble members; Olonscheck et al., 2023). These large ensembles and their ensemble size were chosen based on the availability of sea ice variables. Using multiple large ensembles offers a robust comparison of forced responses and internal climate variability across models (Deser et al., 2020).

The primary sea ice output used in this study is the Arctic SIE, labeled as *siextentn* in the CMIP6 output. In cases where *siextentn* was unavailable, we computed the SIE time series using the SIC data, labeled as *siconc* in the CMIP6 output. SIE is calculated as the total area of all grid cells where SIC exceeds 15%. SIE is a commonly used metric for model comparisons (Shu et al., 2020; Watts et al., 2021; Shen et al., 2021) and our choice of SIE metrics aligns with the existing definitions of RILEs to maintain consistency (Auclair and Tremblay, 2018). However, it is important to note that a limitation of SIE compared to sea ice area (SIA), as highlighted by Notz (2014), is its strong dependency on grid resolution. Additionally, changes in SIA can occur with relatively little change in SIE, which suggests that RILEs defined in terms of SIA may represent fundamentally different processes than those defined using SIE. Nonetheless, we find that our conclusions using SIE are generally consistent with results using SIA (results not shown). We also analyzed SIV, labeled as *sivoln* in the CMIP6 output. If SIV was not available, we computed total Arctic SIV from sea ice thickness (SIT), labeled as *sivol* (grid cell-averaged ice thickness) or *sithick* (sea ice thickness averaged over the ice-covered portion of a grid cell) in the CMIP6 output. When only *sithick* was provided, we calculated SIV by multiplying *sithick* by SIC and the grid cell area. By taking into account the vertical dimension, the SIV metric offers a more comprehensive representation of the condition of the Arctic sea ice cover as it relates more directly to the thermodynamic processes governing its evolution (Stroeve and Notz, 2015).

## 2.2 Model Evaluation

Climate models are powerful tools to analyze the mean state, trends, and variability of the climate system, and how those will evolve into the future. However, the reliability of the conclusions related to sea ice drawn from modeling studies is dependent on the accuracy of the representation of Arctic sea ice and the underlying physical processes embedded within models. To ensure the robustness of our results, we evaluate the performance of the sea ice simulations used in this study using the newly developed SITool (Lin et al., 2021). This tool is designed to assess the skill of CMIP6 simulations by comparing various sea ice metrics with observational references. To do so, we rely on observations of SIE and SIC obtained from the NASA Team (NSIDC-0051) dataset (Cavalieri et al., 1996) and reanalysis of SIT from PIOMAS (Schweiger et al., 2011). Observed SIC and SIT reanalysis data are available since 1979 and, as such, the evaluation of CMIP6 models focuses on the period 1979 to 2014.

SITool reveals noticeable differences between models of the multi-model ensemble in their representation of SIE (Fig. S2); however, the multi-model mean demonstrates a good performance relative to observational data for both March and September (Fig. 1(a)). Additionally, we find that the majority of CMIP6 models effectively replicate the mean and variability of SIC, SIE, and SIT, as well as the spatial distribution of the ice edge (Figs. S2 and S3). Note that some models exhibit larger disparities in one or more metrics when compared to observed references: BCC-CSM2-MR, CAMS-CSM1-0, NESM3, EC-Earth3, and MIROC-E2SL. However, we find that screening out these models does not affect our conclusions (not shown) and, therefore, we have retained this subset of models for the analysis. We also conducted the same evaluation on all members of the five large ensembles for SIE and found that ACCESS-ESM1.5 and MPI-ESM1.2-LR demonstrate better performance in reproducing the mean state, standard deviation, and trend of Arctic sea ice extent, while CanESM5, EC-Earth3, and MIROC6 show slightly lower performance (Figs. 1(b) and S4). Specifically, CanESM5 exhibits a particularly negative trend during 1979

to 2014 compared to observations and MIROC6 shows a less negative trend compared to observations (Fig. S5) and greatly underestimates SIE from November to June (Bianco et al. (2024); Tatebe et al. (2019)).

## 2.3 Definition of RILEs

Several definitions exist in the scientific literature for RILEs in the Arctic, each emphasizing distinct criteria and temporal characteristics (Holland et al., 2008; Lawrence et al., 2008; Döscher and Koenigk, 2013; Paquin et al., 2013; Auclair and Tremblay, 2018; Mioduszewski et al., 2019; Rieke et al., 2023). Holland et al. (2006) used the rate of change exceeding a specific threshold, determined through the derivative of the 5-year mean smoothed time series of SIE. Based on this definition, a RILE is identified when sea ice loss surpasses 0.5 million $km^2$ per year, with the event's duration based on the period during which SIE decreases by more than 0.15 million $km^2$ per year. In contrast, Auclair and Tremblay (2018) defined rapid sea ice declines based on a period lasting at least 4 years, with the trend in the 5-year running mean minimum SIE consistently lower or equal to $-0.3$ million $km^2$ per year. Döscher and Koenigk (2013) characterized a RILE as a drop in summer SIE exceeding 1.2 million $km^2$. According to their definition, a RILE can manifest itself as a single large drop ("one-step event") or a series of up to three consecutive steps involving smaller year-to-year drops ("multi-year event"). Finally, Rieke et al. (2023) assessed rapid ice change events in the Barents Sea using 5-year linear trends of winter (November–April) SIA, using the criteria of trends exceeding two standard deviations of the distribution of 5-year trends in the Community Earth System Model Large Ensemble (CESM-LE) between 2007 and 2025.

For this study, we use the definition from Auclair and Tremblay (2018), for which a RILE is a period lasting at least 4 years, during which the trend in the 5-year running mean minimum SIE is lower or equal to $-0.3$ million $km^2$ per year. We chose this definition as it emphasizes the total amount of loss during a RILE, with the 5-year running mean filtering out interannual variability, as well as limits RILEs to events that last several years rather than single year events, thus focusing on events having a larger impact on climate, ecosystems, and society. We apply the definition from Auclair and Tremblay (2018) to all months of the year maintaining the same threshold. According to this definition, a RILE is even more extreme than the most rapid observed sea ice loss to date. Indeed, over the period 1979–2024, the observed SIE in the Arctic decreased by 0.037 million $km^2$ per year in March and by 0.078 million $km^2$ per year in September (Fetterer et al., 2017), with the most rapid sea ice decline in September over the period 2001-2008 reaching $-0.28$ million $km^2$ per year.

# 3 Results

## 3.1 Seasonality of RILEs

When examining the occurrence of RILEs throughout the year from 1970–2099, we find a distinct regime difference between the first and last six months of the year (Figs. 2 and 3). Indeed, the characteristics of RILEs (e.g., total number of RILEs simulated, duration over multiple years, and intra-seasonal consistency over several months during one year) are noticeably different between winter/spring and summer/fall RILEs, both under the high and low warming scenarios (Fig. 2). From January to June,

very few RILEs are simulated by the CMIP6 multi-model ensemble between 1970 and 2050, with an increasing frequency toward the end of the 21[st] century under high warming scenario (SSP5-8.5) in about a third of the models (Fig. 2(a),(b)). These winter and spring RILEs also exhibit intra-seasonal consistency, meaning that they extend over multiple months of the same year (see darker colors in Fig. 2(a),(b)). For the low warming scenario (SSP1-2.6), only a few RILEs are simulated over the 130 years of our study period, indicating a large contribution from scenario uncertainty on the probability of occurrence of future winter and spring RILEs (Figs. 2(e),(f) and 4(a)). In contrast, between July and December, RILEs are more abundant, though more short-lived, and appear to be randomly distributed throughout the time period when sea ice is present (1970 to consistently ice-free conditions; Jahn et al. (2024); Senftleben et al. (2020)) for both warming scenarios (Fig. 2(c),(d),(g),(h)). We also see a smaller impact of the choice of future scenario on RILEs occurring in the last six months of the year, especially for summer RILEs (Jul-Aug-Sep; Fig. 2(c),(g)). This suggests that forcing factors predominantly influence winter and spring conditions, with little to no role on summer/autumn conditions.

This regime difference between winter and summer RILEs is also present in the large ensembles (Fig. 3). Additionally, we see a large contribution of model uncertainty on the probability of occurrence of future winter RILEs (Fig. 3(a),(b),(c),(d),(e)), something that is also apparent in the CMIP6 multi-model ensemble under high warming scenario (Fig. 2(a),(b)). Model uncertainty is reflected in the timing when winter RILEs first occur as well as their intra-seasonal consistency for multiple months of the year (Fig. 3). In Sect. 3.3, we take a closer look at some important characteristics of RILEs that will shed light on the physical processes leading to this model uncertainty.

While the overall pattern reveals an increase in RILE occurrence from late spring through winter, differences emerge across models (Fig. 5(a)). The CanESM5 large ensemble displays a relatively uniform distribution of RILEs throughout the year, with an average number of RILEs per simulation ranging from 2 in March/April to 2.7 in October. In contrast, the EC-Earth3, ACCESS-ESM1.5, MIROC6 and MPI-ESM1.2-LR large ensembles exhibit more pronounced seasonal variability, with a higher occurrence of RILEs from late spring to early winter. RILEs seasonal patterns for EC-Earth3 and ACCESS-ESM1.5 large ensembles resemble that of the CMIP6 multi-model ensemble for the SSP5-8.5 scenario (Fig. 4(a)). On the other hand, the MIROC6 and MPI-ESM1.2-LR large ensembles exhibit a seasonality pattern in RILE similar to the CMIP6 SSP1-2.6 multi-model distribution, even though all large ensembles analysis here are based on the SSP5-8.5 scenario. The occurrence of RILEs in MIROC6, being similar to RILE occurrence in the multi-model ensemble under the SSP1-2.6 scenario despite the stronger forcing of the SSP5-8.5 scenario, can be attributed to the relatively weak long-term SIE trend in MIROC6, as shown in Fig. S5. However, the comparison between ACCESS-ESM1.5 and MPI-ESM1.2-LR further underscores the complexity: while SIE in both models show similarly weak SIE trends, they differ in their RILES seasonality. This suggests that, while the long-term SIE trend plays a role in determining the seasonality of RILE occurrence, other factors—such as the mean state and internal variability—are also important. For instance, SIE in ACCESS-ESM1.5 has higher internal variability than MPI-ESM1.2-LR but a similar mean state (Fig. 6), which likely contributes to the differences in their seasonal distributions.

Because of the expected increase in sea ice variability as the thickness of the ice cover decreases (Holland et al., 2008) as well as the extreme sea ice loss associated with RILEs, one could expect an early transition toward consistently ice-free conditions in models that simulate many RILEs. However, we find no clear relationship between RILE occurrence and the

timing of consistently September ice-free conditions, with instances of September ice-free conditions occurring at the end of multiple, few, or no RILEs at all. Indeed, some models from the five large ensembles simulate many RILEs before reaching consistently September ice-free conditions (e.g., CanESM5, EC-Earth3, and ACCESS-ESM1.5), while others simulate only a few if any (e.g., MIROC6 and MPI-ESM1.2-LR; Fig. 3). We also find that some models start simulating winter RILEs immediately after the occurrence of consistently ice-free conditions in September (e.g., CanESM5), while for other models (e.g., ACCESS-ESM1.5, MIROC6, EC-Earth3), there is a lag of around 20–30 years between the timing of consistently ice-free conditions in September and the onset of winter RILEs. Additionally, some models simulate a large number of winter RILEs across all ensemble members (CanESM5, ACCESS-ESM1.5, and EC-Earth3), whereas only a few winter RILEs are simulated for MPI-ESM1.2-LR and no RILE is simulated in March for MIROC6 (Fig. 5(a)). This confirms that uncertainty regarding the initiation of winter RILEs is large and strongly model-dependent in addition to the choice of future scenario, as discussed above.

According to the CMIP6 multi-model ensemble, around 50% of the SRILEs end at a SIE value under 2 million $km^2$ and 30% between 0 and 1 million $km^2$ (i.e., consistently ice-free conditions) for both warming scenarios (Fig. 8(d)). This is also the case for the large ensembles: 18–37% of the SRILES end below the 1 million $km^2$ threshold (Fig. 9(d)). For MIROC6, the majority of SRILES have a SIE at around 2.5 million$km^2$ at their onset, which is the lowest value compared to other large ensembles (Fig. 9(b)). Accordingly, 37% of MIROC6 SRILES end at a SIE below 1 million $km^2$ (Fig. 9(d)). While the timing of consistently September ice-free conditions shows little correlation with the occurrence of RILEs, long-lasting RILEs in September (SRILEs) can directly lead to ice-free conditions, although such events are relatively rare. Specifically, RILEs lasting more than 10 years frequently result in ice-free conditions, but these extended events account for less than 15% of all RILEs across both the multi-model ensembles and large ensembles. Indeed, most SRILEs typically persist for 4 to 6 years (Figs. 4(b) and 5(b)). Moreover, this pattern appears consistent across models, suggesting that SRILE duration is not strongly model-dependent.

## 3.2 Probability of occurrence of RILEs

The probability of having at least one SRILE (September RILE) over the period 1970–2099 in the CMIP6 multi-model ensemble is 92% for both scenarios, with a maximum of five SRILEs projected during this period for one single simulation (Fig. 4(c)). When looking at results from the large ensembles, we find disparities across models in terms of the probability of occurrence of SRILEs. There is a 78% probability of having at least one SRILE per simulation with the MIROC6 large ensemble, with 46% of simulations having only one RILE over the period 1970 to 2099. In contrast, the EC-Earth3 large ensemble shows a 100% probability of having at least one SRILE over the same period, with 90% of simulations projected to have more than one SRILE (Fig. 5(c)). This range of results may be related to differences in SIE mean state, variability, and trends among models (Figs. 1 and 6). This again highlights the important role of model uncertainty and the models' mean state on the probability of occurrence of RILEs.

The percentage of simulations exhibiting a RILE before 2030 was analyzed with the multi-model ensemble under both scenarios (SSP1-2.6 and SSP5-8.5) and large ensembles for the SSP5-8.5 scenario (Fig. 4(d) and 5(d)). Approximately 60%

of simulations show a SRILE before 2030, with inter-model differences in the probability. MIROC6 shows a minimum of 26%, while CanESM5 reaches 92%, highlighting strong inter-model variability. This large range of probabilities across models shows that large sea ice model spread remains a concern for CMIP6 models, and that analyzing multiple models is crucial to best characterize the uncertainty inherent in current sea ice projections. While systematic biases in CMIP6 models remain a concern — models can reproduce current sea ice trends for incorrect levels of global warming, as shown by Rosenblum and

Eisenman (2017) for CMIP5 models — our results provide insights by relying on a multi-model ensemble of 26 models and five large ensembles. As the forcings for the SSP5-8.5 and SSP1-2.6 scenarios remain comparable until 2030, the probability of RILEs occurring before 2030 is similar across multiple models under both SSP5-8.5 and SSP1-2.6 scenarios. The analysis of large ensembles reveals that models with high SRILE occurrences before 2030 (80–92%; i.e., CanESM5, ACCESS-ESM1.5 and EC-Earth3) also exhibit increased variability in sea ice extent starting in the late 2010s (Fig. 6(a)). This enhanced vari-

ability increases the likelihood of RILEs before 2030. In contrast, models with lower variability (MIROC6 and MPI-ESM1-2) and an underestimated mean sea ice extent in March (Fig. 1) show a lower (26–30%) probability of SRILE occurrence before 2030. While the different SIE interannual variability in models influences the probability of RILEs, their occurrence remains more frequent in summer than in winter, especially from August to October, stabilizing around 60% in the multi-model ensemble (Fig. 4(d)). Outside the summer season, this probability decreases sharply but does not drop to 0% for the multi-model

ensemble, indicating that RILEs, although less frequent, could still occur before 2030 during other months of the year as well. However, there is a clear model dependence in the seasonal distribution of RILEs. For instance, MIROC6 does not project any RILEs before 2030 outside the summer months, suggesting a strong seasonal confinement in this model.

Interestingly, we find an increased probability of SRILE occurrence after a period of no sea ice loss (i.e., a 10-year period with a neutral or positive SIE trend; Fig. 7). Indeed, while the overall probability of a RILE occurring in the multi-model

ensemble from 2015 until consistently ice-free conditions under SSP5-8.5 is 7%, the likelihood increases to around 20% following a period of no sea ice loss. This increase in probability after a decade of neutral or positive trends is consistent across the CMIP6 multi-model ensemble (Fig. 7) and the five large ensembles (Fig. S7), and those differences are all highly significant (z score > 5). In addition, the SIV trend during the 10-year period of no trend or positive trend does not increase the probability of a RILE occurring in the subsequent years (not shown)

**3.3 Mean State Influence on RILE Occurrence**

SRILEs start occurring in the late 20[th] century and early 2000s for the CMIP6 multi-model ensemble (Fig. 8(a)). By 2025, 50% of SRILEs have already occurred and by 2070 all events have taken place for both scenarios. For the large ensembles, the initiation of SRILEs is similar to the CMIP6 multi-model ensemble: 50% of the total number of SRILEs have already occurred at the earliest by year 2020, as in CanESM5, and at the latest by 2040 for models such as MIROC6 and MPI-ESM1.2-

255 LR (Fig. 9(a)). In these models, the timing of SRILEs is coherent with an increase in September SIE variability (Holland et al., 2008): sea ice variability in CanESM5 starts to increase around 2010 and peaks around 2025, whereas the peak in SIE variability for MIROC6 and MPI-ESM1.2-LR occurs about 20 years later (Fig. 6(a)).

Even though the timing of SRILEs varies by up to two decades across models, the peak of probability of RILEs' onset as a function of SIE is quite consistent for both the CMIP6 multi-model ensemble and large ensembles at slightly less than 4 million $km^2$ (Figs. 8(b) and 9(b)), which suggests that the dependence of RILEs' onset on the mean state is similar across models. This is also the SIE at which the large ensembles simulates the largest values of September sea ice variability (Fig. 6). EC-Earth3 and CanESM5 show a double peak distribution for SIE at the end of SRILEs (Fig. 9(b),(d)), consistent with early and late RILEs due to high sea ice variability for EC-Earth3 and the early increase in variability (2010–2015) for CanESM5 (Fig. 6).

In contrast to September, March RILEs (MRILEs hereafter) occurrences are mostly simulated after 2050. The peak of MRILE occurrence is around 2075 and for SIE values around 11 million $km^2$ in the CMIP6 multi-model ensemble following the high warming scenario (Fig. 8(a),(b)). Among the large ensembles simulating MRILEs (i.e., EC-Earth3, CanESM5, and ACCESS-ESM1.5), the mean SIE at the onset of MRILEs is similar to the CMIP6 multi-model ensemble, except for ACCESS-ESM1.5 for which the peak in MRILE occurrences falls around 15 million $km^2$ (Fig. 9(b)). Uncertainty in the timing of MRILEs is more pronounced across these large ensembles (Fig. 9(a)), highlighting once more the large contribution of model uncertainty on winter RILEs. This uncertainty can be explained by differences in sea ice mean state and/or variability. For example, although the onset years for winter RILEs differ between CanESM5 and EC-Earth3 (Fig. 9(a)), both large ensembles show an increase in sea ice variability as the March SIE falls below 10 million $km^2$ (Fig. 6). In contrast, MPI-ESM1.2-LR exhibits much lower March sea ice variability both as a function of time and SIE (Fig. 6), resulting in few winter RILEs (Fig. 5(a)). Both ACCESS-ESM1.5 and MIROC6 show an increase in sea ice variability over the last few decades of the 21$^{st}$ century as they reach lower SIE (Fig. 6), resulting in an increase in the occurrence of winter RILEs at that time (Fig. 3). The probability density functions of SIE at end of RILEs for March RILEs are flatter and wider than the ones for SRILEs (Figs. 8(d) and 9(d)), which is explained by the large contribution of model uncertainty on the initial SIE as well as on the duration and intra-seasonal consistency of MRILEs.

A peak of RILE occurrences between 5000 and 7500 $km^3$ of SIV is evident for both September and March (Fig. 8(c)) in the CMIP6 multi-model ensemble. This parity in probability of initial SIV between September and March suggests that the average ice volume state may serve as a preconditioning for RILE occurrences. Indeed, we find that more than 20% of SRILEs initiate at a SIV ranging from 4000 to 6000 $km^3$. The declining mean state of SIV likely drives the similarity in SIV at the onset of SRILEs and MRILEs. By 2060–2080, reduced winter SIV (mean: $9.75 \times 10^3$ $km^3$ in March) approaches early 21st-century summer values (mean: $8.23 \times 10^3$ $km^3$ in September 2000–2020), indicating that future winters will resemble today's summers and contribute to RILEs in all seasons (Fig. S8). Additionally, SIV variability differs between these periods: March SIV in 2060–2080 shows lower interannual variability (mean std $\approx 1.5 \times 10^3$ $km^3$) than September SIV in 2000–2020 (mean std $\approx 2.0 \times 10^3$ $km^3$, range 0.5–4.5 $\times 10^3$ $km^3$). This greater summer variability suggests that occasional high summer SIV values in the early 21st century can match low winter SIV levels in the mid-to-late 21st century.

It is important to note that reaching a critical sea ice state is not a sufficient condition for winter RILEs to occur. MIROC6 simulates no winter RILEs before the last two decades of the 21$^{st}$ century (Fig. 3(d)) despite showing a less extensive and thinner winter ice cover (Figs. 1(b) and S6(b)) Holland et al. (2008) showed that, in addition to a forcing perturbation, an

adequately thin ice cover, is necessary to initiate RILEs in September, and our results suggest that this finding is also applicable to winter RILEs, except for EC-Earth3. Indeed, EC-Earth3 simulates winter RILEs at the end of the 20$^{th}$ century (Fig. 3(e)) without meeting the thin sea ice condition.

## 4 Discussion

The increase in RILE occurrence follows the increase in SIE variability, echoing previous findings regarding the influence of large interannual SIE fluctuations on abrupt Arctic SIE declines (Holland et al., 2008; Goosse et al., 2009). This variability is highest when approaching consistently ice-free conditions (Swart et al., 2015; Mioduszewski et al., 2019) with increased SIE variability in summer and autumn attributed to the higher efficiency of open water formation, while variability in November–January is influenced by ice growth (Mioduszewski et al., 2019). Increased variability in SIE has been found to be linked to declining ice thickness (Holland et al., 2008), particularly in winter, with a complex interplay between climate, ice thickness, and geographical factors (Goosse et al., 2009; Mioduszewski et al., 2019). Our results also suggest a preconditioning role of SIV on RILEs, as similar SIV values are observed at the onset of SRILEs and MRILEs. At first, this may seem surprising since winter SIV is generally expected to be larger than summer SIV (e.g., as shown in PIOMAS time series). However, this similarity can be explained by two factors. First, there is large interannual variability in September SIV, so that anomalously high summer SIV values can occasionally match mid-to-late 21$^{st}$ century winter SIV values. Second, March RILEs occur later in the 21$^{st}$ century, when March SIV has declined to levels comparable to late 20$^{th}$-century September SIV. Both interpretations influence the preconditioning role of the SIV, but the declining mean state of sea ice volume seems to be the dominant factor. However, while the total SIV may reach similar values, sea ice spatial distribution will differ. Present-day summer sea ice consists of thicker, multi-year ice in a small area (north of the Canadian Arctic Archipelago and Greenland—where ice survives the summer melt), whereas mid-to-late century winter sea ice will likely be thinner with first-year sea ice covering most of the Arctic Ocean. These differences imply distinct responses to events that could trigger RILEs. According to Döscher and Koenigk (2013), RILEs are controlled by the initial SIV at the onset of the melting period and by the onset of specific atmospheric circulation patterns during summer months. In summer months, thinning ice cover in climate models increases ice extent variability, making it more vulnerable to natural variations and amplifying changes due to the surface albedo feedback resulting in an increased probability of extreme events such as RILEs.

Based on the analyzed CMIP6 model simulations, our study suggests that the most probable occurrence of a SRILE would be in the mid-2020s, or once we reach a September SIE and SIV mean state of approximately 3.5 million $\text{km}^2$ and 6000 $\text{km}^3$, respectively. By comparison, the observed September mean SIE over the past five years (2020–2024) was approximately 4.52 million $\text{km}^2$ (Fetterer et al., 2017), while the September mean SIV over the past five years was approximately 4600 $\text{km}^3$ (Schweiger et al., 2011). As such, the current sea ice state (SIE and SIV) is close to the projected characteristics of SRILEs in CMIP6 simulations. Additionally, the increased probability of SRILE occurrence after a period of no sea ice loss (Fig. 7) suggests that the probability of seeing a RILE in the near future following the recent weak negative trend of observed SIE over 2015–2024 ($-0.017$ million $\text{km}^2$ per year; Fetterer et al. (2017)) is increased, echoing previously proposed ideas of slowing

down as an early warning signal for abrupt climate change (Dakos et al., 2008). This convergence emphasizes the increasing urgency to understand the variability, causes, and impacts of such events. It is important to note, however, that the exact timing and mean state associated with the occurrence of a RILE in the future is still uncertain due to the large contribution of internal climate variability. A deeper examination of the physical mechanisms driving decadal SIE variability in model simulations, through the study of RILEs in the future, is therefore crucial to enhance our capacity to understand and predict the evolution of Arctic sea ice in the coming years and decades.

## 5   Conclusions

Rapid ice loss events (RILEs) were first identified by Holland et al. (2006) and, even though several follow-up studies have been conducted (Lawrence et al., 2008; Holland et al., 2008; Döscher and Koenigk, 2013; Paquin et al., 2013; Auclair and Tremblay, 2018; Mioduszewski et al., 2019; Rieke et al., 2023), we still currently lack a comprehensive overview of the properties of RILEs. Previous studies used a limited number of climate models or mainly focused on one season. Our study assessed RILEs in the Arctic in the past and future during all months of the year using a large set of realizations from five large ensembles and simulations from 26 models participating in CMIP6. The findings illustrate the complex variability of the Arctic sea ice cover through the study of RILEs, shifting from a relatively stable condition in the 20<sup>th</sup> century to a more unpredictable state as we progress further into the 21<sup>st</sup> century, especially during summer. Below, we provide a summary of key results:

– Under a high emission scenario (SSP5-8.5), RILEs occur in the CMIP6 multi-model ensemble (26 models) and five large ensembles for all months of the year (Figs. 2 and 3). All five large ensembles have at least one ensemble member exhibiting a RILE in every month of the year, except for MIROC6 in April. The percentage of CMIP6 models experiencing at least one RILE varies depending on the month of the year, ranging from 62% (February to May) to 96% (August and November).

– The large number of RILEs simulated by the CMIP6 multi-model ensemble in August, September, and October is similar across both future scenarios but that is not the case over the rest of the year (Nov–Jul), with significantly fewer RILEs when using the low emission scenario (SSP1-2.6; Fig. 4(a)). This suggests that the choice of forcing has little influence on the probability of occurrence of end-of-summer RILEs but plays a dominant role in all other months of the year.

– RILEs in winter last longer than in summer and tend to occur toward the end of the century under SSP5-8.5, while they are almost non-existent toward the end of the century in climate projections using SSP1-2.6 (Fig. 2). During the last six months of the year, RILEs are more randomly spread than during the first 6 months of the year, indicating a regime difference between different times of the year. Additionally, it seems that there is a larger influence of model uncertainty on the timing of RILEs in winter.

– The increase in RILE occurrence is driven by the increase in SIE variability (Fig. 6), as already highlighted by Holland et al. (2008) in their analysis of rapid September sea ice retreat. Our results suggest that this finding is also applicable

to winter RILEs. On top of that, the greater the sea ice extent variability is in a model, the more the model simulates RILEs. This result is most clearly illustrated by EC-Earth3 (large ensemble), in which there is a high probability (90%) of experiencing more than one RILE (2–5) during the period from 1970 to consistently ice-free conditions.

– SIV values at the onset of RILEs are similar for both March and September RILEs and both scenarios in the CMIP6 multi-model analysis (Fig. 8(c)). This suggests a preconditioning role of SIV for RILEs with a threshold ranging from 5000 to 7500 $km^3$.

– There is an increase in the probability that a RILE will occur after a 10-year long steady SIE phase (Fig. 7). This increases the probability of a RILE in the near future, following the weak negative SIE trend period between 2015 and 365  2024 (-0.017 million $km^2$ $year^{-1}$)

To conclude, RILEs can happen in any month of the year, not just during summer, depending on future emission scenarios. Given how frequently RILEs occur in climate models, the rapid loss of sea ice that a RILE entails should not come as a surprise if it were to happen in reality.

*Data availability.*  The data from all CMIP6 models are openly available and can be found on the Earth System Grid Federation (ESGF)
nodes: https://esgf-node.llnl.gov/search/CMIP6/. The DOI (Digital Object Identifier) for each model simulation can be found in (Table 2). The PIOMAS (Schweiger et al., 2011) sea-ice volume data can be accessed via the Polar Science Center of the University of Washington: http://psc.apl.uw.edu/research/projects/arctic-sea-ice-volume-anomaly. The observational SIE data from the National Snow and Ice Data Center (NSIDC) can be accessed via https://nsidc.org/data/nsidc-0192/versions/3.

*Author contributions.*  AS, FM, TF and AJ conceptualized the science plan. AS performed the analyses, produced the figures and wrote the
manuscript based on the insights from the co-authors. AS, PDR, AJ, FM, DD, and TF contributed to writing, reviewing, and editing. Initial analysis on the seasonality of RILEs and the SIE threshold where RILEs begin to occur were performed in a subset of large ensemble by DQ, ES and MO, respectively, under guidance from CWP and AJ.

*Competing interests.*  The authors declare that they have no conflict of interest.

*Acknowledgements.*  Annelies Sticker and Patricia DeRepentigny are funded by the European Union (ERC, ArcticWATCH, 101040858).
Views and opinions expressed are however those of the author(s) only and do not necessarily reflect those of the European Union or the European Research Council Executive Agency. Neither the European Union nor the granting authority can be held responsible for them. François Massonnet is a F.R.S.-FNRS Research Associate. David Docquier is funded by BELSPO through the RESIST project (contract no. RT/23/RESIST). Computational resources have been provided by the supercomputing facilities of the Université catholique de Louvain

(CISM/UCL) and the Consortium des Équipements de Calcul Intensif en Fédération Wallonie Bruxelles (CÉCI) funded by the Fond de la Recherche Scientifique de Belgique (F.R.S.–FNRS) under convention 2.5020.11. The contributions of A. Jahn, C. Wyburn-Powell, D. Quint, E. Shivers, and M. Ortiz were supported by NSF-CAREER award 1847398. We acknowledge the World Climate Research Programme's Working Group on Coupled Modelling, which is responsible for CMIP, and we thank the climate modeling groups for producing and making available their model output. We acknowledge the use of ChatGPT (https://chat.openai.com/) to improve the writing style of a few paragraphs.

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

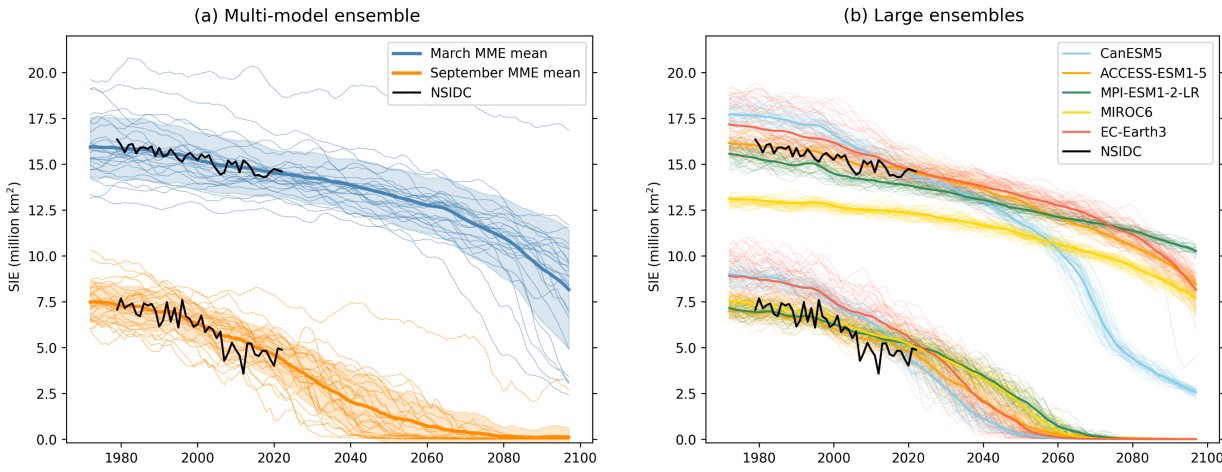

**Figure 1.** March (top lines) and September (bottom lines) 5-year running mean SIE evolution over the historical period and the high emission scenario SSP5-8.5 for (a) the CMIP6 multi-model ensemble (26 models, 1 member per model), with thin lines representing individual models, thick lines the multi-model ensemble mean, and shaded areas one standard deviation across the multi-model ensemble, and (b) five large ensembles with thin lines representing individual ensemble members and thick lines the ensemble mean. The black lines show the observations from NSIDC.

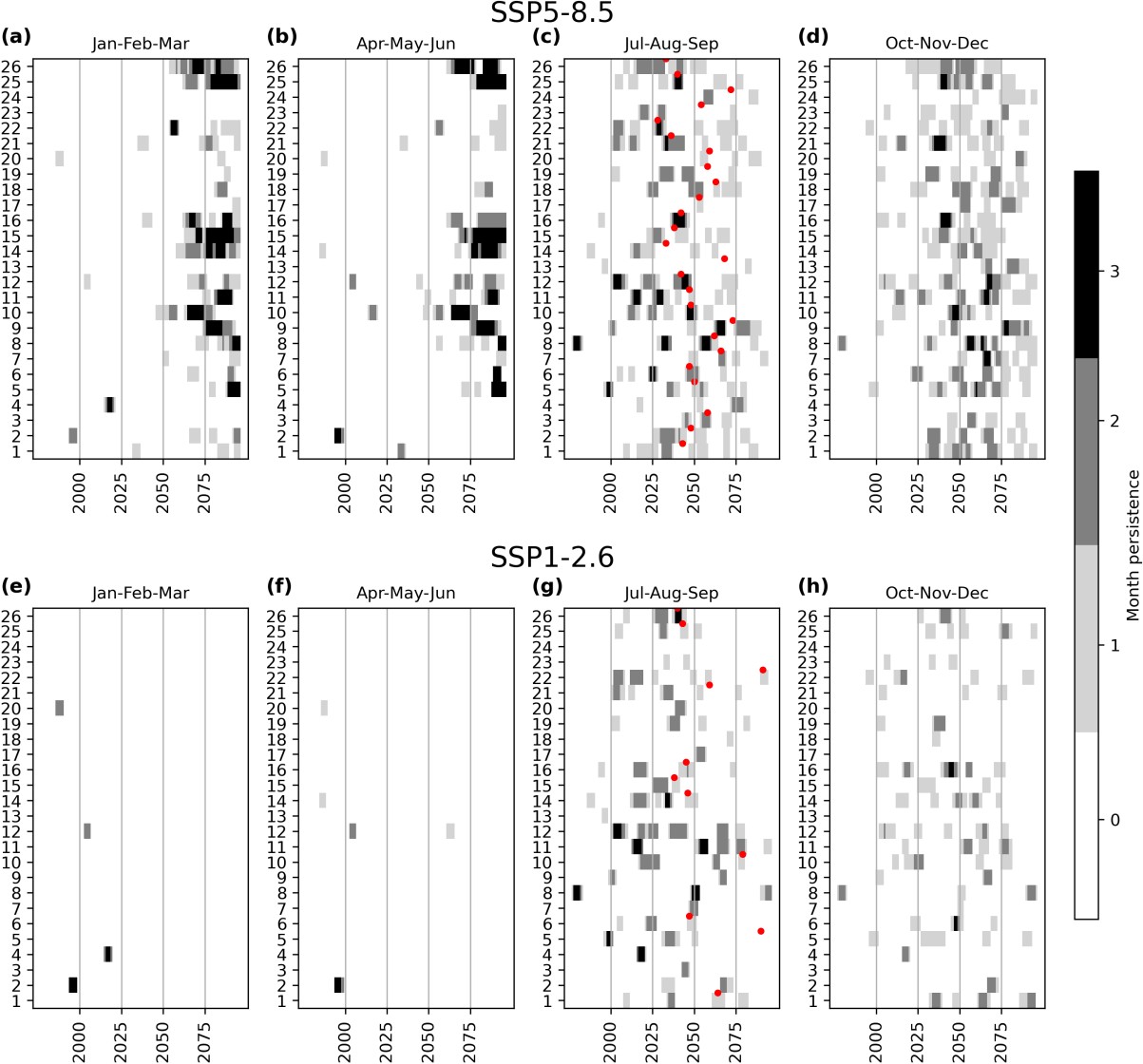

**Figure 2.** Occurrences of RILEs from 1970–2099 in the first ensemble member of the 26 different CMIP6 models following the SSP5-8.5 (top row) and SSP1-2.6 (bottom row) scenarios. Each panel shows a period of three months, with light grey representing RILEs occurring over one of the three months, dark grey representing RILEs occurring over two of the three months, and black representing RILEs occurring over all three months of the season. The numbers on the y-axis refer to the 26 different models listed in Table 1. Red dots in the Jul-Aug-Sep panels indicate the first year of consistently ice-free conditions in September (i.e., first year of five consecutive years when the smoothed September SIE falls below 1 million $km^2$).

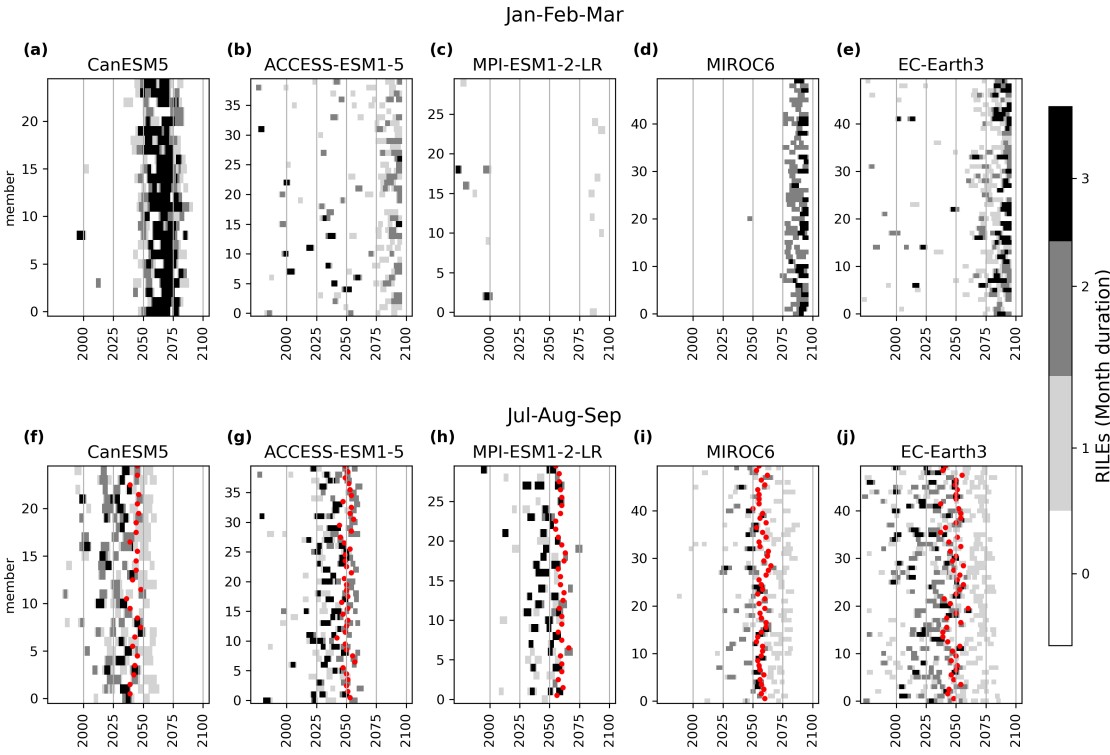

**Figure 3.** Same as in Fig. 2 but for the periods Jan-Feb-Mar (top row) and Jul-Aug-Sep (bottom row) in the five large ensembles following the SSP5-8.5 scenario.

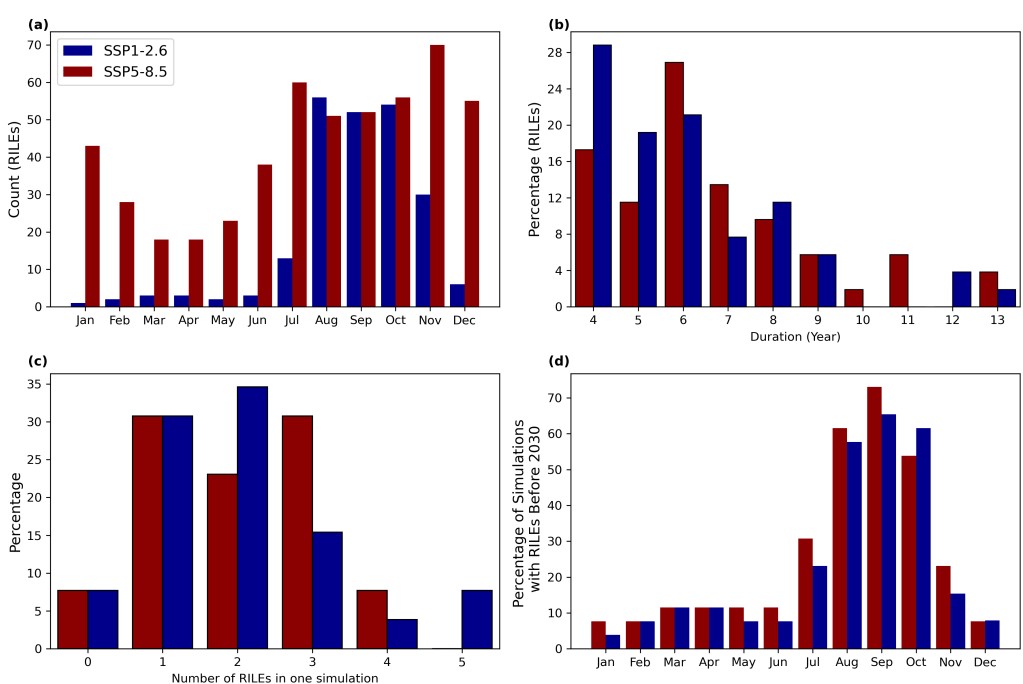

**Figure 4.** RILE characteristics in the CMIP6 multi-model ensemble: (a) total number of RILEs per month, (b) percentage of SRILEs as a function of their duration in years, and (c) percentage of SRILEs per simulation, and (d) percentage of simulations having at least one RILE occurrence before 2030 in each month for the CMIP6 multi-model ensemble over 1970–2099 under the high (red) and low (blue) warming scenarios.

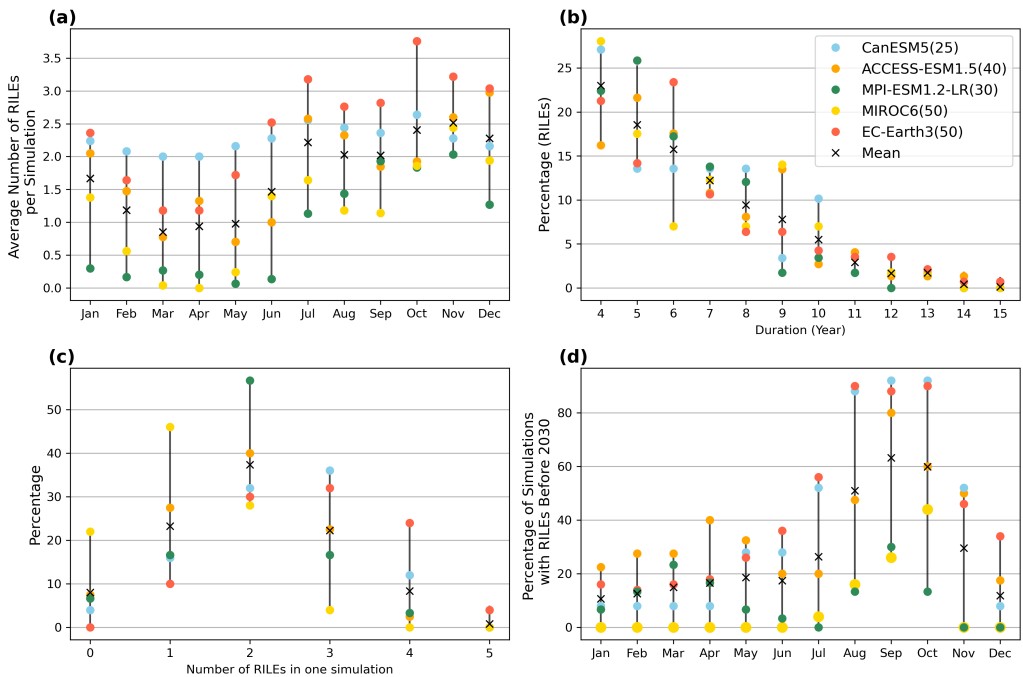

**Figure 5.** RILE characteristics in five large ensembles following the high SSP5-8.5 scenario: (a) average number of RILEs per simulation per month, (b)-(d) Same as Fig. 4. The black X represents the mean across the five large ensembles, and the numbers in parentheses in the legend indicates the ensemble size for each large ensemble.

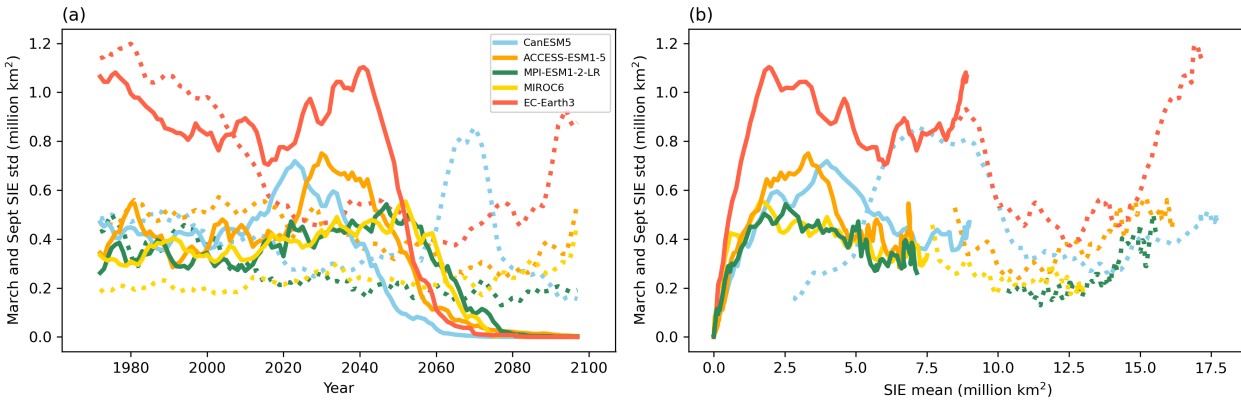

**Figure 6.** Standard deviation of the 5-year running mean SIE for March (dotted lines) and September (solid lines) as a function of (a) time and (b) September and March SIE mean state for the five large ensembles following the high emission scenario SSP5-8.5.

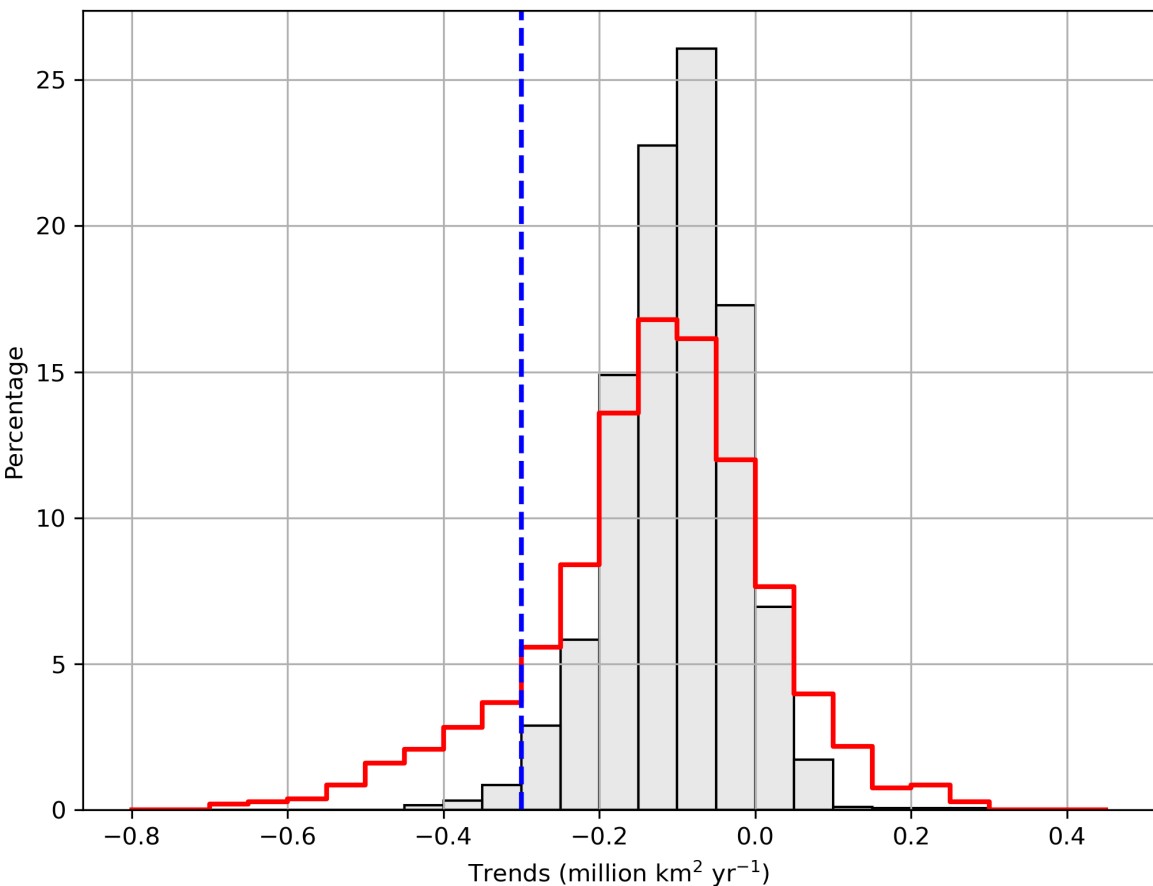

**Figure 7.** Distribution of all possible 10-year Arctic September SIE trends (grey) and trends after a period of stability (red) for the CMIP6 multi-model ensemble from 2015 to consistently ice-free conditions using the SSP5-8.5 scenario. A period of stability is defined as a 10-year period with a neutral or positive SIE trend. The 10-year trends are computed on the 5-year running mean SIE timeseries. The dotted blue line indicates the threshold used to define a RILE (Sect. 2.3 for more details). Fig. S7 shows similar results but for the five large ensembles.

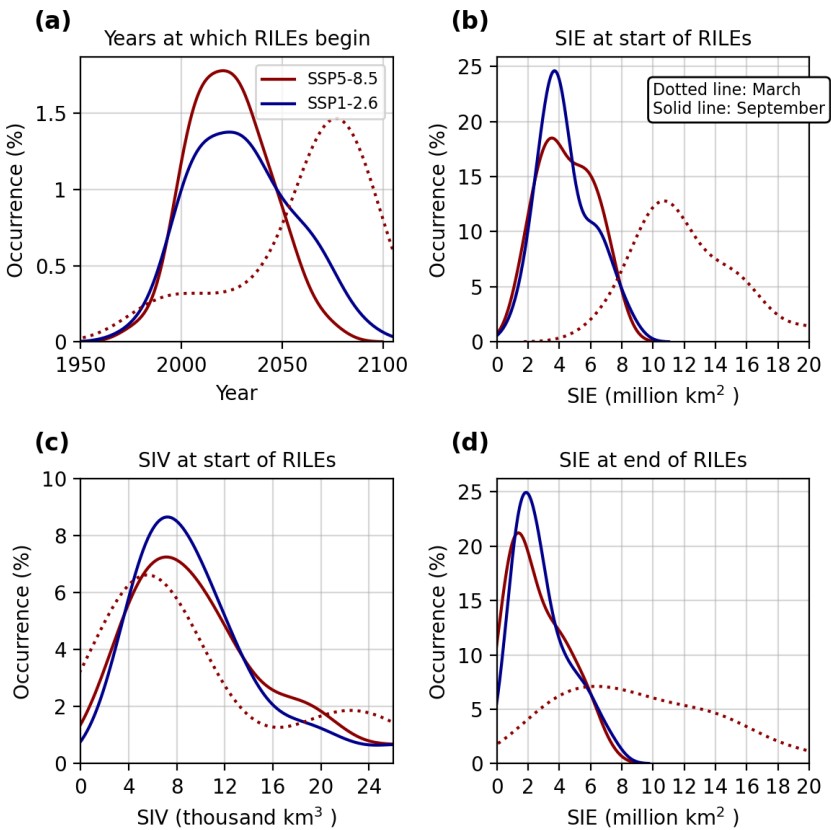

**Figure 8.** RILE characteristics in the CMIP6 multi-model ensemble: probability density function of the (a) years, (b) SIE and (c) SIV at which RILEs begin, and (d) the SIE at which RILEs end for September (solid lines) and March (dotted lines) for the CMIP6 multi-model ensemble over 1970–2099 under the high (red) and low (blue) warming scenarios. Note that we do not show results for the low warming scenario in March due to very few RILEs being simulated.

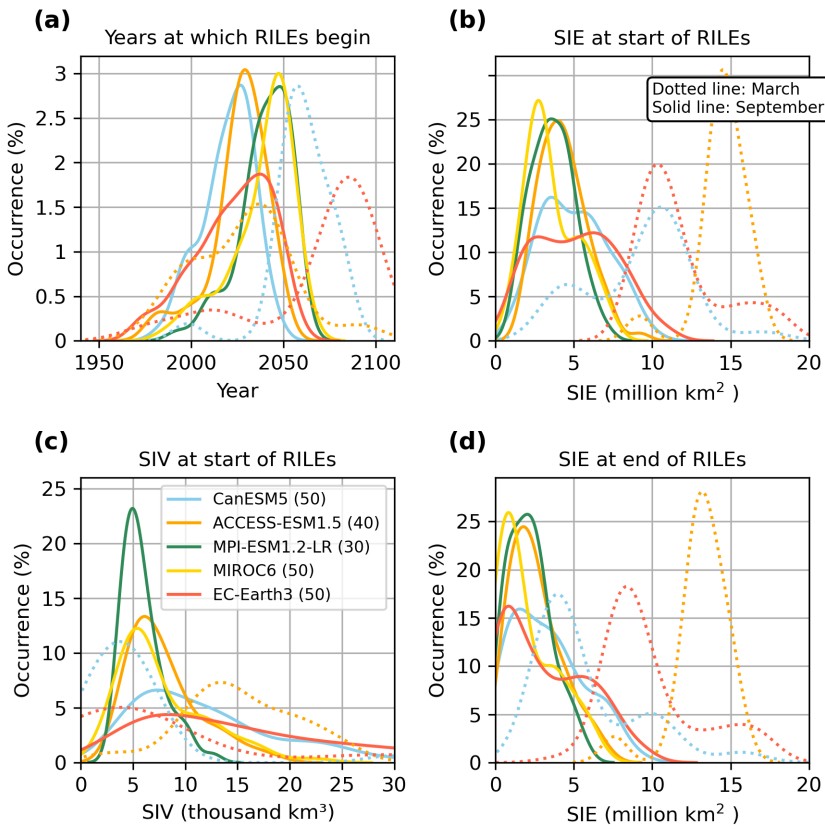

**Figure 9.** Same as Fig. 8 but for the five large ensembles following the high emission scenario SSP5-8.5. The numbers in parentheses in the legend indicates the ensemble size for each large ensemble. Note that we do not show results for MPI-ESM1.2-LR and MIROC6 in March due to very few MRILEs being simulated in these two models.

**Table 1.** Ocean and sea ice components, together with their nominal resolution, of the CMIP6 models used in this analysis. The resolution corresponds to the ocean/sea ice component. The sea ice component for models #19–20 is unnamed but uses the "Semtner zero-layer" thermodynamic and "Hibler 79" dynamic schemes.

| MODEL NAME | OCEAN MODEL | SEA ICE MODEL | OCEAN / SEA ICE RESOLUTION |
|---|---|---|---|
| 1. ACCESS-CM2 | MOM5 | CICE5.1.2 | 100 km |
| 2. ACCESS-ESM1.5 | MOM5 | CICE 4.1 | 100 km |
| 3. BCC-CSM2-MR | MOM4 | SIS2 | 50 km |
| 4. CAMS-CSM1-0 | MOM4 | SIS1.0 | 100 km |
| 5. CESM2-WACCM | POP 2 | CICE 5.1 | 100 km |
| 6. CESM2 | POP 2 | CICE 5.1 | 100 km |
| 7. CNRM-CM6-1-HR | NEMO 3.6 | Gelato 6.1 | 25 km |
| 8. CNRM-CM6-1 | NEMO 3.6 | Gelato 6.1 | 100 km |
| 9. CNRM-ESM2-1 | NEMO 3.6 | Gelato 6.1 | 100 km |
| 10. CanESM5 | NEMO3.4.1 | LIM2 | 100 km |
| 11. EC-Earth3-Veg | NEMO 3.6 | NEMO-LIM3 | 100 km |
| 12. EC-Earth3 | NEMO 3.6 | NEMO-LIM3 | 100 km |
| 13. GFDL-ESM4 | GFDL-OM4p5 | GFDL-SIM4p5 | 50 km |
| 14. HadGEM3-GC31-LL | NEMO-HadGEM3-GO6.0 | CICE-HadGEM3-GSI8 | 100 km |
| 15. HadGEM3-GC31-MM | NEMO-HadGEM3-GO6.0 | CICE-HadGEM3-GSI8 | 25 km |
| 16. IPSL-CM6A-LR | NEMO 3.6 | NEMO-LIM 3 | 100 km |
| 17. MIROC-ES2L | COCO4.9 | COCO4.9 | 100 km |
| 18. MIROC6 | COCO4.9 | COCO4.9 | 100 km |
| 19. MPI-ESM1.2-HR | MPIOMI 1.6.3 | Unnamed | 50 km |
| 20. MPI-ESM1.2-LR | MPIOMI 1.6.3 | Unnamed | 250 km |
| 21. MRI-ESM2-0 | MRI.COM 4.4 | MRI.COM 4.4 | 100 km |
| 22. NESM3 | NEMO v3.4 | CICE4.1 | 100 km |
| 23. NorESM2-LM | MICOM | CICE | 100 km |
| 24. NorESM2-MM | MICOM | CICE | 100 km |
| 25. TaiESM1 | POP2 | CICE4 | 50 km |
| 26. UKESM1-0-LL | NEMO-HadGEM3-GO6.0 | CICE-HadGEM3-GS18 | 100 km |

**Table 2.** References for different CMIP6 simulations under various scenarios used in this study.

| MODEL NAME | ref historical simulations | ref SSP5-8.5 simulations | ref SSP1-2.6 simulations |
|---|---|---|---|
| 1. ACCESS-CM2 | Dix et al. (2019a) | Dix et al. (2019c) | Dix et al. (2019b) |
| 2. ACCESS-ESM1.5 | Ziehn et al. (2019a) | Ziehn et al. (2019c) | Ziehn et al. (2019b) |
| 3. BCC-CSM2-MR | Wu et al. (2018) | Xin et al. (2019b) | Xin et al. (2019a) |
| 4. CAMS-CSM1-0 | Rong (2019c) | Rong (2019b) | Rong (2019a) |
| 5. CESM2-WACCM | Danabasoglu (2019a) | Danabasoglu (2019c) | Danabasoglu (2019b) |
| 6. CESM2 | Danabasoglu (2019d) | Danabasoglu (2019f) | Danabasoglu (2019e) |
| 7. CNRM-CM6-1-HR | Voldoire (2019a) | Voldoire (2019e) | Voldoire (2020) |
| 8. CNRM-CM6-1 | Voldoire (2018) | Voldoire (2019d) | Voldoire (2019b) |
| 9. CNRM-ESM2-1 | Seferian (2018) | Voldoire (2019f) | Voldoire (2019c) |
| 10. CanESM5 | Swart et al. (2019a) | Swart et al. (2019c) | Swart et al. (2019b) |
| 11. EC-Earth3-Veg | EC-Earth-Consortium (2019b) | EC-Earth-Consortium (2019f) | EC-Earth-Consortium (2019d) |
| 12. EC-Earth3 | EC-Earth-Consortium (2019a) | EC-Earth-Consortium (2019e) | EC-Earth-Consortium (2019c) |
| 13. GFDL-ESM4 | Krasting et al. (2018) | John et al. (2018b) | John et al. (2018a) |
| 14. HadGEM3-GC31-LL | Ridley et al. (2019a) | Good (2020b) | Good (2020a) |
| 15. HadGEM3-GC31-MM | Ridley et al. (2019b) | Jackson (2020b) | Jackson (2020a) |
| 16. IPSL-CM6A-LR | Boucher et al. (2018) | Boucher et al. (2019b) | Boucher et al. (2019a) |
| 17. MIROC-ES2L | Hajima et al. (2019) | Tachiiri et al. (2019b) | Tachiiri et al. (2019a) |
| 18. MIROC6 | Tatebe and Watanabe (2018) | Shiogama et al. (2019b) | Shiogama et al. (2019a) |
| 19. MPI-ESM1.2-HR | Jungclaus et al. (2019) | Schupfner et al. (2019a) | Schupfner et al. (2019b) |
| 20. MPI-ESM1.2-LR | Wieners et al. (2019c) | Wieners et al. (2019a) | Wieners et al. (2019b) |
| 21. MRI-ESM2-0 | Yukimoto et al. (2019a) | Yukimoto et al. (2019c) | Yukimoto et al. (2019b) |
| 22. NESM3 | Cao and Wang (2019) | Cao (2019b) | Cao (2019a) |
| 23. NorESM2-LM | Seland et al. (2019a) | Seland et al. (2019c) | Seland et al. (2019b) |
| 24. NorESM2-MM | Bentsen et al. (2019a) | Bentsen et al. (2019c) | Bentsen et al. (2019b) |
| 25. TaiESM1 | Lee and Liang (2020a) | Lee and Liang (2020c) | Lee and Liang (2020b) |
| 26. UKESM1-0-LL | Tang et al. (2019) | Good et al. (2019b) | Good et al. (2019a) |