# Peer review of "Seasonality and scenario dependence of rapid Arctic sea ice loss events in CMIP6 simulations"

_EGUsphere, 2024_

## Author Comment (AC1)

**Author Comments**

**Seasonality and scenario dependence of rapid Arctic sea ice loss events in CMIP6 simulations**

Annelies Sticker, François Massonnet, Thierry Fichefet, Patricia DeRepentigny, Alexandra Jahn, David Docquier, Christopher Wyburn-Powell, Daphne Quint, Erica Shivers, and Makayla Ortiz

Submitted to *The Cryosphere* on 19 Jun 2024, manuscript id: EGUSPHERE-2024-1873

January 21, 2025

**Anonymous Referee #1**

**General comments**

Sticker et al. analyse rapid Arctic sea ice loss events (RILEs)—year-to-year reductions in sea ice extent greatly exceeding that expected from the long-term trend—in CMIP6 and large-ensemble simulations. In contrast to previous studies, they examine RILEs in all seasons and assess the consistency across the CMIP6 ensemble. They find that RILEs do occur in winter/spring but less frequently than in summer/autumn and strongly dependent on future warming scenario. Interestingly, the sea ice volume (SIV) at which RILEs are typically triggered is about the same for March and September RILEs, despite differences in initial sea ice extent (SIE).

This work advances our understanding of large-scale sea ice variability on interannual timescales in the context of RILEs, which are likely to become more frequent in the coming decades as the authors explain and demonstrate. I found the result relating to pre-conditioning based on the initial SIV being similar for winter and summer RILEs particularly striking, although I think the discussion/interpretation of it needs a little expansion (see specific comments).

I see no issues with the methodology overall, except perhaps with using the CMIP6 results to infer to the likelihood of a RILE starting in the next few years. It is not clear where the rather specific probability of 63% stated in the abstract comes from, and I think there are caveats here which I don't see that the authors have acknowledged (see specific comments).

The figures are presented well, and I appreciate the concise length of the manuscript. One weakness in terms of presentation and structure is the results section 3. I found myself having to jump back and forth between Figs. 2–5 too many times while reading the text, making it difficult to follow. I suggest the authors consider rearranging figures (and possibly some of the text) to improve readability here. I have given a few suggestions in the specific comments below.

Otherwise, subject to addressing these points and the other minor and technical points noted below, I believe this work should be published in The Cryosphere.

**Reply.** We thank the reviewer for the helpful feedback on our manuscript. We will clarify the probability of 63% by adding a new figure to the manuscript (see reply to specific comments below) and by adding a paragraph about it in the main text. We will also discuss the role of declining mean SIV and variability in preconditioning RILEs. Additionally, the problem with the structure of the text in Section 3 will be addressed by dividing the figures 4 and 5 into two parts and rearranging the figures. We will:

- separate panels e–g from Figures 4 and 5 and create a new combined figure (Fig. C3 a & b), by including figure Fig C2 (see replies to specific comments).

- rearrange the order of the figures in the text to ensure consistency and logical flow (for instance, we will mention Figure 6 in its appropriate place in Section 3.3, after the discussion of Figure 4a in Section 3.2).

**Specific comments**

**Major comments**

> Preconditioning/SIV result: this result is, at first thought, quite surprising, because one expects the typical SIV in winter to be larger than the typical SIV in summer (e.g., see PIOMAS time series). But it can be explained by either:
>
> Winter RILEs mainly occurring later (i.e., when winter SIV of the mid–late 21st century is presumably comparable to summer SIV of the late 20th century)
>
> Large interannual variability in winter and summer SIV, such that anomalously high summer SIVs are comparable to anomalously low winter SIVs
>
> It seems like both are relevant, from Fig. 2 (for 1) and Fig. S4 (for 2), but the problem is that authors do not mention either (or anything else). Currently they just state that the initial SIV is the same for winter and summer RILEs based on Figs. 4c and 5c, and this therefore indicates a role of preconditioning (e.g., L255, 266, 311). I think these interpretations, and some comment on the extent to which one or the other is dominant, should be added to the discussion.

**Reply.** This is a good point. To further expand the discussion on this point, we will include an additional analysis comparing the mean state and standard deviation of sea ice volume (SIV) for three periods: September (2000–2020), March (2000–2020), and March (2060–2080), using the multi-model ensemble (Fig. C1). The results indicate that winter SIV during the early 21st century is higher than summer SIV for the same period. However, by the mid-to-late 21st century, winter SIV declines significantly, reaching levels comparable to those of summer SIV in the early 21st century (Fig. C1, left panel). Furthermore, the range of interannual variability, which is relatively high for winter SIV in the early 21st century, decreases substantially by the mid-to-late 21st century (Fig. C1, right panel). Consequently, anomalously high summer SIVs due to large interannual variability become comparable to low winter SIVs. Our results suggest that the similarity in SIV at the onset of SRILEs and MRILEs is likely more influenced by the declining mean state of sea ice volume than by variability alone.

**Action.** We will include the following discussion in the main text and figure C1 in Supplementary Materials.

We will include the following paragraph in Section "3.3 Mean State Influence on RILE Occurrence":

"The similarity in SIV at the onset of SRILEs and MRILEs is likely influenced by the declining mean state of sea ice volume. By the mid-to-late 21st century, reduced winter SIV (mean: $9.75 \times 10^3$ km$^3$ for March 2060–2080) approaches early 21st-century summer values (mean: $8.23 \times 10^3$ km$^3$ for September 2000–2020), suggesting that winter conditions will resemble today's summer conditions, contributing to RILEs occurrence in all seasons (Fig. C1). Additionally, the SIV variability is different between early 21st century summer and winter and mid-to-late 21st century winter (Fig. C1). The March SIV during 2060–2080 has relatively low interannual variability (mean std ~$1.5 \times 10^3$ km$^3$), while the September SIV during 2000–2020 shows higher variability (mean std ~$2.0 \times 10^3$ km$^3$) and a wider range of standard deviations going from 0.5 to 4.5 thousand square kilometers. This greater summer variability suggests that anomalously high summer SIV values in the early 21st century can reach values similar to low winter SIV in the mid-to-late 21st century."

We will include the following paragraph in Section "4 Discussion":

"Our results suggest that the SIV plays a preconditioning role in RILEs, as similar SIV values are observed at the onset of SRILEs and MRILEs. At first, this may seem surprising since winter SIV is generally expected to be larger than summer SIV (e.g., as shown in PIOMAS time series). However, this similarity can be explained by two factors. First, there is large interannual variability in September SIV, so that anomalously high summer SIV values can occasionally match mid-to-late 21st century winter SIV values. Second, March RILEs occur later in the 21st century, when March SIV has declined to levels comparable to late 20th-century September SIV. Both interpretations influence the preconditioning role of the SIV, but the declining mean state of sea ice volume seems to be the dominant factor. However, while the total SIV may reach similar values, sea ice spatial distribution will differ. Present-day summer sea ice consists of thicker, multi-year ice in a small area (north of the Canadian Arctic Archipelago and Greenland—where ice survives the summer melt), whereas mid-to-late century winter sea ice will likely be thinner with first-year sea ice covering most of the Arctic Ocean. These differences imply distinct responses to events that could trigger RILEs."

[Figure]

**Figure C1**. *(Left) Boxplots showing the mean Arctic sea ice volume ($10^3 \, km^3$) for September (2000–2020), March (2000–2020) and March (2060–2080) for the multi-model ensemble using the historical simulation (2000-2014) and the warming scenario SSP5-8.5 (2014-2020 and 2060-2080), with means (red dots) and PIOMAS observations (black dots) highlighted.(Right) Boxplots of the interannual variability (standard deviation) of Arctic sea ice volume for the multi-model ensemble for the same periods.*

Chance of RILE by 2030: I'm uncertain about the authors' claim in the abstract that the real Arctic has a "63% chance" of exhibiting a RILE by 2030. Firstly, this value of 63% is only present in the abstract (L19) and so it is not clear where it comes from. In any case, there are surely too many uncertainties with this estimate to state such a specific value, so it would be better to rephrase into a more general statement with approximate likelihood (e.g., "suggest about a 60% chance"). From a readability standpoint, it would also make more sense to put this sentence after the sentence which currently follows (i.e., make it, "The study of RILEs is particularly opportune [...]", then "Based on CMIP6 [...]").

The authors need to explain somewhere in the main text how they are deriving this estimate and mention the underlying assumptions. In particular, they need to note their estimate of the likelihood is going to be affected by the issue of model uncertainty/spread. I suspect their estimate could be biased towards certain models with stronger sea ice declines (e.g., CanESM5, EC-Earth3, from Fig. 1b). For example, EC-Earth3 appears to have quite a large left tail in the distribution of trends after stability (Fig. S6), meaning it probably contributes more to the estimate of imminent RILE likelihood. In a couple of places, the authors use current estimates of the real Arctic sea ice volume to help justify the mapping from CMIP6 simulations to what might happen in the real world (e.g., L253, L272). The problem here is that there remain large systematic biases in Arctic sea ice (trends) in CMIP6 compared to the real world, so even if you match the current trends and/or values of the SIV/SIE, the models may be simulating that with unrealistic global warming, for example (e.g., Rosenblum and Eisenman, 2017). This, then, casts doubt on how applicable the underlying statistics leading to the model estimate of RILE likelihood at a given SIV level/stability period is to the real world with the same SIV level/stability period.

**Reply.** We thank the referee for raising this important point. We agree that the derivation of the 63% likelihood requires clarification, and we will address this in the main text by adding two more panels (Fig. C2) to figure C3&b. The referee also raises an important point about systematic biases in CMIP6 models. Studies like Rosenblum and Eisenman (2017) have indeed shown that models can reproduce current trends in sea ice extent or volume with unrealistic climate forcing. However, our results provide key insights that help contextualize these concerns:

- **Model spread and robustness**: While individual models may exhibit biases, our analysis relies on a broad set of CMIP6 models and large ensembles. The fact that multiple models show a high probability of RILEs increases the robustness of our findings, even if the response to forcings differs across models. However,

using many biased models doesn't make the bias go away, as we know from the results from Rosenblum and Eisenman (2017), especially if they are primarily biased in one direction. So, we agree that we should acknowledge the bias and make it clear that it is a model-based probability.

- **Link with ice volume and variability**: Our results show that RILEs are primarily associated with low SIV and increased SIE interannual variability. This relationship is robust within CMIP6 simulations and aligns with recent observations highlighting the increased vulnerability of sea ice as its thickness and volume decline (Sumata et al., 2023). This reinforces the idea that similar conditions in the real world could also favor RILEs.

[Figure]

**Figure C2**. *Percentage of simulations having at least one RILE occurrence before 2030 in each month and for (left side) the high warming scenario SSP5-8.5 and the low warming emission scenario SSP1-2.6 in the multi-model ensemble and (right side) the high warming scenario for the 5 LE.*

**Action.** We will separate panels e–g from Figures 4 and 5 and create a new combined figure (Fig. C3 a & b) where we will add two new panels (Fig. C2) that show the percentage of simulations having at least one RILE occurrence before 2030 in each month. We will also add the following discussion in Section "3.2 Probability of occurrence of RILEs".

"The percentage of simulations exhibiting a RILE before 2030 was analyzed with the multi-model ensemble under both scenarios (SSP1-2.6 and SSP5-8.5) and large ensembles for the SSP5-8.5 scenario (Fig. C2). The results indicate an approximate 60% probability of observing a RILE in September before 2030, with differences between models. MIROC6 shows a minimum of 26%, while CanESM5 reaches 92%, highlighting strong inter-model

variability. While systematic biases in CMIP6 models remain a concern—models can reproduce current sea ice trends with unrealistic climate forcing (Rosenblum and Eisenman, 2017)—our results provide insights by relying on a multi-model ensemble and 5 large ensembles. Additionally, the probability of RILEs occurring before 2030 is similar across multiple models under both SSP5-8.5 and SSP1-2.6 scenarios, likely because forcings for these scenarios remain comparable until 2030, with more pronounced differences emerging in the mid-to-late 21st century.

The analysis of large ensembles under SSP5-8.5 reveals that models with high SRILE occurrences (80-92%) (e.g., CanESM5, ACCESS-ESM1-5 and EC-Earth3) exhibit increased variability in sea ice extent starting in the late 2010s (Fig. S5). This variability increases the likelihood of RILEs before 2030. In contrast, models with lower variability (MIROC6 and MPI-ESM1-2) and an underestimated mean sea ice extent in March (Fig. 1) project a lower (26-30%) probability of SRILEs occurrence before 2030.

While the different SIE interannual variability in models influences the probability of RILEs, the probability remains high during summer months, especially from August to October, stabilizing around 60% in the multi-model ensemble (Fig. C3b). Outside the summer season, this probability decreases sharply but does not drop to 0% for the ensemble, indicating that RILEs, although less frequent, could still occur early during other times of the year as well.

However, there is a clear model dependence in the seasonal distribution of RILEs. For instance, MIROC6 does not project any RILEs before 2030 outside the summer months, suggesting a strong seasonal confinement in this model. In contrast, models such as MPI-ESM1-2-LR and ACCESS-ESM1-5 exhibit a relatively stable probability of RILEs throughout the year, with little seasonal variability. Conversely, the probability of observing a RILE outside the summer season decreases substantially for CanESM5 and EC-Earth3, highlighting a more pronounced seasonality in these models."

Readability of section 3: this could mostly be addressed by rearranging the figures. Figure 4 panels e–g seem like they should be in a separate figure to panels a–d, and similarly for Fig. 5. Separating the e–g panels out of each and combining into one figure might be better. Panels e–g are referred to in section 3.1 before the a–d panels for both Figs. 4 and 5. Figure 6 is also mentioned in section 3.2 before Fig. 4a, which is first mentioned in section 3.3.

**Reply.** We appreciate the reviewer's feedback regarding the arrangement of the figures to improve the readability of Section 3. We agree that separating panels e–g from Figures 4 and 5 and combining them into a new figure will help streamline the presentation of the results.

**Action.** Specifically, we will:

1. Remove panels e–g from Figures 4 and 5 and create new combined figures (C3a and C3b), by including panels from Fig. C2 (see previous comment).
2. Rearrange the order of the figures in the text to ensure consistency and logical flow. For instance, we will mention Figure 6 in its appropriate place in Section 3.3, after the discussion of Figure 4a in Section 3.2.

[Figure]

**Figure C3a.** *RILEs characteristics in 5 CMIP6 LE: (a) average number of RILEs per simulation, (b) percentage of SRILEs as a function of their duration in years, (c) percentage of SRILEs per simulation, and (d) percentage of simulations having at least one RILE occurrence before 2030 in each month under the high warming scenario SSP5-8.5.*

[Figure]

**Figure C3b.** *RILEs characteristics in the CMIP6 multi-model ensemble: (a) total Number of RILEs, (b) percentage of SRILEs as a function of their duration in years, (c)percentage of SRILEs per simulation, and (d) percentage of simulations having at least one RILE occurrence before 2030 in each month under the high warming scenario SSP5-8.5 (red) and under the low warming scenario SSP1-2.6 (blue).*

**Minor comments**

> L5: I suggest making it clear that you are referring to year-to-year changes in total sea ice extent, as this description could equally apply to sub-seasonal time scale and/or regional scale sea ice loss. Indeed, there is a separate body of literature on such "very rapid ice loss events (VRILEs)", which is obviously quite different to what you study. It is unfortunate that there is such a clash of terminology/acronyms, and while I suspect the term "RILE" will ultimately be more commonly used for the interannual events you are describing, it is better to be clear up front. Simply adding "year-to-year" in front of "reductions" and "total" in front of sea ice extent would be one way to address this point.

**Action.** To address this point, we will add the terms "year-to-year" before "reductions" and "total" before "sea ice extent" to make it clear that we are referring to sub-decadal changes in Arctic sea ice extent.

L58–65: As above, I suggest briefly noting somewhere in this paragraph the distinction from short timescale "very rapid ice loss events" (e.g., McGraw et al., 2022; Wang et al. 2020).

**Action.** To clearly differentiate RILEs from VRILES, we will add the following paragraph about VRILEs in Section "1 Introduction".

"Sea ice loss events are also studied on shorter time scales, with Very Rapid Ice Loss Events (VRILEs) describing abrupt declines in sea ice that happen over days to weeks (Wang et al., 2020; McGraw et al., 2022). VRILEs are often associated with atmospheric and oceanic anomalies that enhance ice loss over short periods, typically within a season. While these studies have deepened our understanding of subseasonal sea ice variability, the focus of the present study is on Rapid Ice Loss Events (RILEs), which manifest on subdecadal to decadal timescales."

L84: Worth noting that Arctic sea ice stabilizes in SSP1-2.6 (e.g., IPCC AR6/TS). So, towards the latter part of the 21st century RILES are occurring in the absence of a background trend for most models.

**Reply.** Looking at SIE evolution with SSP1-2.6 emission scenario (Fig. C4), it seems that there is still a clear background trend in Arctic sea ice extent during the 2020s, particularly in the September projections, when most RILEs are simulated. However, Arctic sea ice indeed stabilizes later in the 21st century under SSP1-2.6 while RILEs are still occurring. We agree that this highlights that RILEs can arise even in the absence of a background trend.

**Action.** We will add Fig. C4 to Supplementary Materials and will update the sentence as following:

"We also employed two sets of climate projections following low and high warming scenarios, specifically SSP1-2.6 and SSP5-8.5, which correspond to a top-of-atmosphere radiative forcing in 2100 of 2.6 and 8.5 $W/m^2$ with respect to pre-industrial levels, respectively (O'Neill et al., 2016). Under SSP1-2.6, the Arctic SIE continues to decline in the earlier decades of the 21st century before stabilizing towards the latter part of the century (after ~2050) (Fig.C3)."

[Figure]

**Figure C4.** March (top lines) and September (bottom lines) 5-year running mean SIE evolution over the historical period and high emission scenario SSP1-2.6 for (a) the CMIP6 multi-model ensemble (26 models, 1 member per model), with thin lines representing individual models, thick lines the multi-model ensemble mean, and shaded areas one standard deviation across the multi-model ensemble, and (b) 5 large ensembles with thin lines representing individual ensemble members and thick lines the ensemble mean. The black lines show the observations from OSI-SAF.

> L100: Since you are considering total sea ice extent/area (SIE/SIA) on interannual time scales, I would not expect model resolution to matter too much. The difference is that a change in SIA can occur with relatively little change in SIE, so that RILEs defined in terms of SIA are (potentially) physically/fundamentally different to those defined in terms of SIE. I think it's fine to use SIE (especially considering you examine SIV too), but might be worth noting this as another reason for checking the impact of using SIA.

**Reply.** This is indeed worth clarifying.

**Action.** We will update the text as follows:

"However, it is important to note that a limitation of SIE compared to sea ice area (SIA), as highlighted by Notz (2014), is its strong dependency on grid resolution. Additionally, changes in SIA can occur with relatively little change in SIE, which suggests that RILEs defined in terms of SIA may represent fundamentally different processes than those defined using SIE. Nonetheless, we find that our conclusions using SIE are generally consistent with results using SIA (results not shown)."

> L126–128: "not shown": it is good that the authors include some model evaluation. While I understand they do not wish to clutter their manuscript with too much tangential material, I think here the authors could include a figure demonstrating this in their supplementary materials, or at least cite some other studies (surely the extreme November–June sea ice departures from observations in MIROC6 have been found already, for example?)

**Reply:** Bianco et al. (2024) look at the variability of Arctic SIE on interannual and multidecadal time scales for 29 models and found that MIROC6 exhibits a low mean SIE bias during 1850–1919. Additionally, Tatebe et al. (2018) reported an underestimation of sea ice extent in MIROC6 and attributed it to slightly more rapid Arctic warming in the model compared to observations. Finally, Fig. 5C shows the SIE mean trends for 1980-2014. From this figure, it can be seen that CanESM5 simulates a too strong negative long-term trend and MIROC6 simulates a too weak long-term trend compared to observations.

**Action:** We will add these two references (Bianco et al. (2024), Tatebe et al. (2018)) to support our discussion and Fig. C5 to the supplementary materials.

> L174: I agree with comparison between and interpretation of Figs. 4e,f,g and 5e,f,g, but would suggest there is a bit more to say here about the (e) panels in particular as not all large ensembles look the same. The CanESM5 large ensemble has a fairly uniform distribution, and looks like CMIP6 multi-model ensemble for SSP5-8.5, whereas the MIROC6 large ensemble's distribution looks more like the CMIP6 distribution for SSP1-2.6, even though SSP5-8.5 simulations are used for all large ensembles in Fig. 5e. If I understand correctly, this is because MIROC6 has a relatively weak long-term trend and so looks like the CMIP6 average for SSP1-2.6 even when simulating SSP5-8.5. Either way, I think the interpretation should be described in the text (rather than just, "the characteristics are found to be very similar").

**Reply.** We thank the referee for raising the lack of interpretation here. We will update panels e,f,g of figure 5 (Fig. C3a). The updated figure will now show the percentage of the average number of RILEs per simulation (a), the duration of RILEs (b), the number of RILEs in one simulation (c), and the percentage of simulations with RILEs before 2030 (d) for each model, highlighting the variability between models in a clearer way.

**Action.** We will add Fig. C5 to Supplementary Materials as well as the following discussion in Section "3.1 Seasonality of RILEs":

"The first panels of Figures C3a and C3b illustrate the seasonality of RILES for the five large ensembles and a multi-model ensemble under the two SSP scenarios. While the overall pattern reveals an increase in RILES occurrence from late spring through winter, differences emerge across models. The CanESM5 large ensemble displays a relatively uniform distribution of RILES throughout the year. In contrast, the EC-Earth3 and ACCESS-ESM1-5 large ensembles exhibit more pronounced seasonal variability, with a higher occurrence of RILES from late spring to early winter. Their seasonal patterns resemble that of the CMIP6 multi-model ensemble for SSP5-8.5. On the other hand, the MIROC6 and MPI-ESM1-2-LR large ensembles exhibit a seasonality pattern similar to the CMIP6 SSP1-2.6 distribution, even though all large ensembles analysis here are based on the SSP5-8.5 scenario.

The occurrence of RILEs in MIROC6, being similar to RILEs occurrence in the multi-model ensemble under the SSP1-2.6 scenario despite the stronger forcing of SSP5-8.5, can be attributed to the relatively weak long-term SIE trend in MIROC6, as shown in Fig. C5. However, the comparison between ACCESS-ESM1-5 and MPI-ESM1-2-LR further underscores the complexity: while SIE in both models show similarly weak SIE trends, they differ in their RILES seasonality. This suggests that, while the long-term SIE trend plays a role in determining the seasonality of RILES occurrence, other factors—such as the mean state and internal variability—are also important. For instance, SIE in ACCESS-ESM1-5 has higher internal variability than MPI-ESM1-2-LR but a similar mean state, which likely contributes to the differences in their seasonal distributions."

[Figure]

**Figure C5.** The 1979-2014 monthly anomalies of Arctic sea ice extent (million km2) from the observational NSIDC-0051 (solid black) and the members of the 5 LE from CMIP6 with

the mean trend in dashed lines. The standard deviation (Std, million km2) and trend (million km2 decade−1) of the monthly anomalies of observational ice extent are displayed.

> L178: I think this also follows (more obviously?) from Fig. 3 rather than Fig. 5 (see also point above about readability in this section)

**Reply.** This is not clear in Fig.3 as the grouping into three-month periods makes it difficult to distinguish monthly patterns. This could be addressed if we rearrange the panels as suggested in a previous comment.

**Action.** We will ensure that the readability of the figures is enhanced for better interpretation.

> L185: I see you have defined "consistently September ice free" in the caption of Fig. 2, but this should be stated in the main text (either here or somewhere in Section 2). I also think a reference should be provided for this (e.g., Senftleben et al., 2020).

**Reply.** This is stated in Section 3, L164 with one reference. We can also add Senftleben et al. (2020), thanks for the suggestion.

**Action.** The reference Senftleben et al., 2020 will be added.

> L211: "No SRILE occurs after consistently ice-free conditions occur in September": is this not obvious? You need sea ice in September to have a September RILE. Is this what you meant to write?

**Reply.** We want to inform the reader that a rebound of sea ice can sometimes occur after a RILE, even if the Arctic Ocean reaches consistently ice-free conditions in September. However, we agree that the relationship between sea ice in September and September RILEs is generally implicit.

**Action.** We will remove the sentence.

**Technical corrections**

> L29: "more vulnerable to atmospheric and oceanic variability": "variability" → "forcing" (or "more vulnerable to variability in atmospheric and ocean forcing")?

**Action.** We will replace "variability" by "forcing".

> L32: "a interannual" → "an interannual"

**Action.** We will replace "a interannual" by "an interannual" in line 32.

> L49: "during one or several" → "during one or over several"

**Action.** We will replace "during one or several" by "during one or over several" in line 49.

> L140: I suggest not introducing the acronym "RICE" here, since it is only used in this line and nowhere else in the rest of the manuscript.

**Reply.** The acronym RICE might be confusing as the events described are specific to a region and season.

**Action**. To address this, we will remove the acronym that was only used here.

> L220: "onset to" → "onset on"
>
> L255: "preconditionning" → "preconditioning"
>
> L264: "had" → "has"

**Action.** We will make the necessary corrections.

> L270: repeated reference (I think it's clear that you are still describing results from Döscher and Koenigk, 2013, cited on L266).

**Action.** We will remove the citation.

> L272: "in mid-2020s" → "in the mid-2020s"

**Action.** We will replace "in mid-2020s" by "in the mid-2020s".

> L295: Unclear; re-phrase? (E.g., to "...the percentage of members with at least one RILE per year ranges from 62–96%, and every model experiences at least one RILE during the analysis period")

**Action.** We will clarify the sentence as follows:

"For the CMIP6 multi-model ensemble, the percentage of models experiencing at least one RILE varies depending on the month of the year, ranging from 62% in the month with the fewest models simulating a RILE to 96% in the month with the most models simulating a RILE. Notably, every model experiences at least one RILE during the analysis period."

L306: "ice" → "sea ice"

**Action.** We will replace "ice" by "sea ice" in line 306.

L308: Remove or rephrase "nicely" (e.g., "This result is most clearly illustrated by EC-Earth3, for example, ...")

**Action.** We will replace "nicely" by "most clearly" in line 308.

L320: I commend the authors for including the specific data citations for all CMIP6 models and simulations—a lot of other studies, particularly multi-model studies, do not bother with these. However, I do suggest moving them from the supplementary materials to the main text (e.g., add them to Table 1?). Otherwise, I don't think that cross referencing systems will detect the citations, which is important for tracking usage of CMIP data (and it would then have been a waste of your time to include them in the first place!)

**Action.** We will add citation in the table.

L322: This NSIDC dataset has a proper citation with DOI; I suggest adding this rather than a URL.

**Action.** We will add the citation.

Figure 1, caption: "NISDC" → "NSIDC"

**Action.** We will make the correction.

Figure 6, caption: "using SSP5-8.5 scenario" → "using the SSP5-8.5 scenario"

**Action.** We will make the correction.

**References**

McGraw, M. C., Blanchard-Wrigglesworth, E., Clancy, R. P., and Bitz, C. M.: Understanding the Forecast Skill of Rapid Arctic Sea Ice Loss on Subseasonal Time Scales, J. Climate, 35, 1179–1196, https://doi.org/10.1175/JCLI-D-21-0301.1, 2022

Rosenblum, E. and Eisenman, I.: Sea Ice Trends in Climate Models Only Accurate in Runs with Biased Global Warming, J. Climate, 30, 6265–6278, https://doi.org/10.1175/JCLI-D-16-0455.1, 2017

Senftleben, D., Lauer, A., and Karpechko, A.: Constraining Uncertainties in CMIP5 Projections of September Arctic Sea Ice Extent with Observations, J. Climate, 33, 1487–1503, https://doi.org/10.1175/JCLI-D-19-0075.1, 2020

Wang, Z., Walsh, J., Szymborski, S., and Peng, M.: Rapid Arctic Sea Ice Loss on the Synoptic Time Scale and Related Atmospheric Circulation Anomalies, J. Climate, 33, 1597–1617, https://doi.org/10.1175/JCLI-D-19-0528.1, 2020

Bianco, E., E. Blanchard-Wrigglesworth, S. Materia, P. Ruggieri, D. Iovino, and S. Masina, 2024: CMIP6 Models Underestimate Arctic Sea Ice Loss during the Early Twentieth-Century Warming, despite Simulating Large Low-Frequency Sea Ice Variability. *J. Climate*, **37**, 6305–6321, https://doi.org/10.1175/JCLI-D-23-0647.1.

Tatebe, Hiroaki & Ogura, Tomoo & Nitta, Tomoko & Komuro, Yoshiki & Ogochi, Koji & Takemura, Toshihiko & Sudo, Kengo & Sekiguchi, Miho & Abe, Manabu & Saito, Fuyuki & Chikira, Minoru & Watanabe, Shingo & Mori, Masato & Hirota, Nagio & Kawatani, Yoshio & Mochizuki, Takashi & Yoshimura, Kei & Takata, Kumiko & O'ishi, Ryouta & Kimoto, Masahide. (2018). Description and basic evaluation of simulated mean state, internal variability, and climate sensitivity in MIROC6. Geoscientific Model Development Discussions. 1-92. 10.5194/gmd-2018-155.
* * *
REFERENCES

Bianco, E., E. Blanchard-Wrigglesworth, S. Materia, P. Ruggieri, D. Iovino, and S. Masina, 2024: CMIP6 Models Underestimate Arctic Sea Ice Loss during the Early Twentieth-Century Warming, despite Simulating Large Low-Frequency Sea Ice Variability. *J. Climate*, **37**, 6305–6321, https://doi.org/10.1175/JCLI-D-23-0647.1.

Tatebe, Hiroaki & Ogura, Tomoo & Nitta, Tomoko & Komuro, Yoshiki & Ogochi, Koji & Takemura, Toshihiko & Sudo, Kengo & Sekiguchi, Miho & Abe, Manabu & Saito, Fuyuki & Chikira, Minoru & Watanabe, Shingo & Mori, Masato & Hirota, Nagio & Kawatani, Yoshio & Mochizuki, Takashi & Yoshimura, Kei & Takata, Kumiko & O'ishi, Ryouta &

Kimoto, Masahide. (2018). Description and basic evaluation of simulated mean state, internal variability, and climate sensitivity in MIROC6. Geoscientific Model Development Discussions. 1-92. 10.5194/gmd-2018-155.

---

## Author Comment (AC2)

**Author Comments**

**Seasonality and scenario dependence of rapid Arctic sea ice loss events in CMIP6 simulations**

Annelies Sticker, François Massonnet, Thierry Fichefet, Patricia DeRepentigny, Alexandra Jahn, David Docquier, Christopher Wyburn-Powell, Daphne Quint, Erica Shivers, and Makayla Ortiz

Submitted to *The Cryosphere* on 19 Jun 2024, manuscript id: EGUSPHERE-2024-1873

January 21, 2025

**Anonymous Referee #2**

The authors of this study investigate Arctic Rapid Ice Loss Events (RILEs) in the CMIP6 ensemble, including their frequency through the year, dependence on emission scenario, and possible preconditioning conditions. The manuscript is well written, the figures high quality, and the methodology sound and well described. I think this paper will be an important contribution to the literature and have only minor suggestions before it should be published.

**Reply.** We thank the reviewer for his/her constructive comments on the manuscript and we provide our replies below for specific comments.

**Specific Comments:**

> Variability results: (Line 259 and elsewhere). The impact of variability on the frequency or probability of RILEs is a major result, yet there are no figures showing variability. The value of the large ensembles in this analysis is the ability to look at variability for a single model and how that relates to RILEs, something you can't do with the CMIP6 multi-model ensemble. A figure like 5b and 5c but with variability at RILE start would be beneficial. Additionally, Figure S5 may be relevant for the main manuscript and something you could add as a panel to Figure 1. I think you should spend a bit more time on this result since in your conclusions you list it (Line 307-309) but no main figures show larger variability in a model leads to more RILES.

**Reply.** Thank you for highlighting the importance of variability in relation to RILE frequency and probability. We agree that variability is a key result, and we already include discussion about variability in section 3.3 (lines 215,216,220,225, and 235-240). While a new figure illustrating SIE variability at the onset of RILEs may not capture its importance, we agree that Fig. S5 provides valuable insights and is relevant in the main manuscript.

**Action.** We will add Fig. S5 to the main text as this figure is referred several times in Section ' 3.3 Mean State Influence on RILE Occurrence '.

> Figure 4: The order of how you refer to the panels in the text is confusing and you should consider re-ordering them.

**Reply.** We appreciate the reviewer's feedback regarding the arrangement of the figures to improve the readability of Section 3, which was also raised by reviewer 1. We agree that separating panels e–g from Figures 4 and 5 and combining them into a new figure will help streamline the presentation of the results better.

**Action.** We will:

1. Remove panels e–g from Figures 4 and 5 and create a new combined figure (Fig.C3a and b).
2. Rearrange the order of the figures in the text to ensure consistency and logical flow. For instance, we will mention Figure 6 in its appropriate place in Section 3.3, after the discussion of Figure 4a in Section 3.2.

[Figure]

**Figure C3a.** *RILEs characteristics in 5 CMIP6 LE: (a) average number of RILEs per simulation, (b) percentage of SRILEs as a function of their duration in years, (c) percentage of SRILEs per simulation, and (d) percentage of simulations having at least one RILE occurrence before 2030 in each month under the high warming scenario SSP5-8.5.*

[Figure]

**Figure C3b.** *RILEs characteristics in the CMIP6 multi-model ensemble: (a) total Number of RILEs, (b) percentage of SRILEs as a function of their duration in years, (c)percentage of SRILEs per simulation, and (d) percentage of simulations having at least one RILE occurrence before 2030 in each month under the high warming scenario SSP5-8.5 (red) and under the low warming scenario SSP1-2.6 (blue).*

Figure 5: I do not understand what 5f and 5g are showing and 5f is not referenced in text anywhere. Maybe this figure or panels from this figure could be in the supplementary material.

**Reply.** We thank the reviewer for pointing out the lack of clarity and lack of interpretation. However, we think that these results should stay in the main manuscript.  We will update Figure 5 by removing panels e-g and creating a new combined figure that more clearly presents the characteristics of RILEs (Fig. C3a). The updated figure will show the percentage of the average number of RILEs per simulation (a), the duration of RILEs (b), the number of RILEs in one simulation (c), and the percentage of simulations with RILEs before 2030 (d) for each model, highlighting the variability between models in a clearer way. Panel b) of Figure C3a represents the percentage of SRILEs as a function of their duration, while Panel c) shows the percentage of simulations as a function of the number of SRILEs occurring in one simulation.

**Action.** We propose to add the following discussion in Section "3.2 Probability of occurrence of RILEs":

"SRILEs most commonly last between 4 and 6 years, although some events can persist for up to 15 years (Fig. C3a, b)). Notably, when SRILEs extend beyond 10 years, they often directly lead to ice-free conditions. Additionally, these results appear to be consistent across models, indicating that the duration of SRILEs is not strongly model-dependent.

The maximum number of SRILEs observed in a single simulation is 5, which only occurs in the EC-Earth3 model—known for its large internal sea ice varability (Fig. C3a, c)). For most simulations, 2 SRILEs are the most likely outcome. Interestingly, while a majority of simulations feature at least one SRILE, the results show significant model disparities: EC-Earth3 predicts SRILEs in 100% of its simulations, consistent with its high variability. In contrast, MIROC6 simulations mostly show only 1 SRILE or, in many cases, no SRILEs at all, highlighting a weaker tendency for rapid sea ice loss events in this model."

Figure 6: If RILEs initiate more often after a period of stable SIE trends, does this imply that the SIV would still have a negative trend during this period. So the ice is thinning but not changing extent? Can you add some text about what's going on with SIV during these periods?

**Reply.** We appreciate the question regarding SIV behavior during periods when SIE trends are stable, particularly in relation to the initiation of RILEs. To address this, we analyze the SIV trends during these periods (Fig. C6). Specifically, we investigate whether periods of stability in SIE coincide with "hidden sea ice loss," characterized by a decline in SIV despite the absence of shrinking in SIE. Such conditions may create a more vulnerable state for the sea ice, potentially increasing the likelihood of RILEs in subsequent years. We identify all periods of SIE stability accompanied by combined SIV loss and analyze the SIE trends following these periods. Our results indicate that there is no significant increase in the frequency of RILEs after such "hidden sea ice loss" periods, with only ~12% of subsequent SIE trends falling below -0.3 million $km^2$ per year (Fig. C6, yellow histogram). This result provides further insight into the preconditions for RILE initiation: a period of stable or positive 10-year SIE trends combined with sea ice volume loss does not appear to increase the probability of a RILE occurring in the subsequent years.

**Action.** We will keep Fig. 6 as it is in the main text and add the following sentence in Section "3.2 Probability of occurrence of RILEs":

"Furthermore, these periods of no sea ice loss combined with a decline of SIV for the same 10-year period does not appear to increase the probability of a RILE occurring in the subsequent years (not shown)."

[Figure]

**Figure C6.** Distribution of all possible 10-year Arctic September SIE trends (grey) for the CMIP6 multi-model ensemble under the SSP5-8.5 scenario, from 2015 to consistently ice-free conditions. The trends are computed using a 5-year running mean of the SIE time series. The red outline represents SIE trends following periods of stability, where stability is defined as a 10-year period with zero or positive SIE trends. The yellow outline corresponds to SIE trends following periods where 10-year SIE trends are equal to or greater than zero, combined with 10-year negative trends in SIV. The blue dashed line indicates the threshold of -0.3 million km$^2$/yr, used to define RILEs (see Section 2.3 for details).

Line 2 and Line 8: "practically ice free" and "nearly ice free" is awkward. Just define "ice free" and go with it.

**Reply.** We believe it is important to retain "practically" in this context, as the definition used here refers to an ice-free Arctic as 1 million km$^2$ of sea ice or less, rather than the complete disappearance of sea ice. This choice reflects the diversity of "ice-free Arctic" definitions in literature (see Jahn et al., 2024).

**Action.** We will replace "nearly" with "practically" in the sentence in line 8 to provide greater clarity.

> Line 6-7: This is a confusing sentence because if you're looking from start of satellite era, why are you listing 2000-2008 rates?

**Reply.** We are listing 2001-2008 because it is the largest observed rate since the beginning of the satellite observing period.

**Action.** To clarify this sentence, we will rephrase it as follows:

"The extreme sea ice loss associated with RILEs in climate models exceeds any observed rates of sea ice loss since the start of the satellite era, including the largest observed rate of -0.28 million km2 per year during 2001–2008."

> Line 7: what is "it"? Maybe say "As such, there could be a much faster transition..."

**Action.** We will rephrase "As such, it could lead to a much faster transition…" in "As such, there could be a much faster transition…"

> Lines 101-104: This description of these conversions is confusing. Maybe use equations instead of text? Also, shouldn't grid cell area come in here?

**Reply.** Thank you for pointing this out. We agree that the original description was too brief and could cause confusion. We will not use equations to maintain consistency with the rest of the text but will ensure that the methodology is precise.

**Action.** We will clarify the text to better describe how we computed sea ice volume (SIV) when it was not directly available, explicitly including the role of grid cell area in these calculations. We will update the text as follows:

"We also analyzed SIV, labeled as sivoln in the CMIP6 output. If SIV was not available, we computed it from sea ice thickness (SIT), labeled as sivol (grid cell-averaged ice thickness) or sithick (sea ice thickness averaged over the ice-covered portion of a grid cell) in the CMIP6 output.  When only sithick was provided, we calculated the SIV by multiplying sithick by sea ice concentration (SIC) and the grid cell area."

> Line 138-139: What do you mean a RILE can manifest as one year event or several years? This sentence sounds like you mean the metric by the Döscher and Koenigk since that's the last you discussed, and that's a one-year definition.

**Reply.** We are referring to Döscher and Koenigk's definition.

**Action.** We will replace "As such," by "According to their definition," in lines 138-139.

> Line 140: Is a RICE substantively different than a RILE? Why not include this in the other metrics described?

**Reply.** The acronym RICE might be confusing as the events described are specific to a region and season.

**Action.** To address this, we will remove the acronym that was only used here.

> Line 203: What is "a period"? Is it a single year where the 10-year previous trend was positive or zero? This sentence needs clarification.

**Action.** We will clarify to" period of no sea ice loss" to match the previous definition (i.e., a 10-year period with a zero or positive SIE trend)

> Line 295: The sentence where you say the percentage goes from 62-96% is confusing and should be clarified.

**Action.** We will rephrase the sentence as follows:

"For the CMIP6 multi-model ensemble, the percentage of models experiencing at least one RILE varies depending on the month of the year, ranging from 62% in the month with the fewest models simulating a RILE to 96% in the month with the most models simulating a RILE. Notably, every model shows at least one RILE during the analysis period."

---

## Referee Report (RR1)

**Sticker et al.: Seasonality and scenario dependence of rapid Arctic sea ice loss events in CMIP6 simulations**

I have read through the author response to both my comments and the other referee comments. I think the authors have done a great job addressing my minor concerns by highlighting variability, assessing "hidden sea ice loss", and modifying figures. Overall, with these changes, I find the already well-written manuscript is clearer and the results are important. Well done, authors, and thank you for your diligent work and responses! I recommend this paper for publication.